EMBO
Molecular Medicine

# Peptidomic analysis of CSF reveals new biomarker candidates for amyotrophic lateral sclerosis

Besnik Muqaku[1], Johannes Dorst[1,2], Maximilian Wiesenfarth[2], Markus Otto ![ORCID][3], Albert C Ludolph[1,2] & Patrick Oeckl ![ORCID][1,2✉]

## Abstract

**Amyotrophic lateral sclerosis (ALS) is a devastating neurodegenerative disease, and novel biomarkers are needed. We applied mass-spectrometry-based peptidomic analysis in cerebrospinal fluid (CSF) samples of ALS and non-neurodegenerative control patients (Con) from a discovery ($n = 48$) and validation ($n = 109$) cohort for biomarker discovery. Systematic selection revealed a panel of eight novel peptide biomarker candidates for ALS (out of 33,605) derived from seven proteins. In the validation cohort, NFL, MAP1B, MYL1, and APOC1 peptides were upregulated, and peptides from CADM3, SCG1, and PENK were downregulated in ALS compared to Con. The peptides (except NFL) were not changed in other neurodegenerative diseases, including Alzheimer´s disease, frontotemporal dementia and Parkinson´s disease. Combination of all peptides in a logistic regression model led to an area under the curve value of 98% for the discrimination of ALS from controls. Data of the NFL peptide strongly correlated with an established NFL immunoassay (Ella, $r = 0.97$). The peptide biomarker candidates are derived from proteins with different function, and their determination with our method provides the opportunity for simultaneous investigation of key processes in ALS.**

**Keywords** ALS; Peptidomics; Biomarker; Neurodegeneration; Muscle
**Subject Categories** Biomarkers; Neuroscience; Proteomics

## Introduction

Amyotrophic lateral sclerosis (ALS) is a devastating and progressive neurodegenerative disease that affects motor neurons, leading to neuronal degeneration, muscle weakness, and atrophy (Hardiman et al, 2017). The exact pathomechanisms of ALS remain unknown for the majority of sporadic ALS (sALS) patients, whereas up to 10% of ALS patients have a genetic form of ALS (gALS) with mutations in the *C9orf72* and *SOD1* gene as the most frequent ones (Hardiman et al, 2017).

ALS diagnosis is based on clinical symptoms (Brooks et al, 2000; Ludolph et al, 2015), hampering the identification of patients in the preclinical stage and requiring diagnosis at specialized centers. Recently, a novel disease-modifying therapy for ALS was approved for SOD1 mutation carriers (Miller et al, 2022). There is an urgent need for biomarkers since they are an important tool to improve the diagnostic procedure and support the development of novel drug candidates. Fluid biomarkers, measured in blood (plasma and serum) or cerebrospinal fluid (CSF), could ideally facilitate diagnosis at a presymptomatic disease stage, support the evaluation of disease progression and detect early treatment effects (Feldman et al, 2022). They could also be beneficial for patient stratification during enrollment in clinical trials.

Neurofilament light chain (NFL) is the most promising and well-characterized biomarker in ALS, showing increased concentration in blood and CSF possibly due to axonal neurodegeneration (Feneberg et al, 2018; Steinacker et al, 2016). It is already used to support clinical diagnosis of ALS and has successfully been used as a read-out in the SOD1 antisense oligonucleotide trial (tofersen) (Miller et al, 2022). However, NFL concentration seems only to increase a few weeks or months before disease onset (Benatar et al, 2018; Feldman et al, 2022; Weydt et al, 2016), it is not specific to ALS (Ashton et al, 2021), and does not change during disease progression (Lu et al, 2015; Steinacker et al, 2017).

Peptides play a critical role in cellular communication, regulation, and signaling processes, and are involved in various physiological processes, from immune responses to neural activity. Hormones and neuropeptides are the most studied bioactive peptides (Foreman et al, 2021; Phetsanthad et al, 2023). Their synthesis into active forms usually requires the intervention of several proteolytic enzymes (Bergeron et al, 2000; Cawley et al, 2012). In addition, post-translational modification (PTM) may be necessary. The proteolytic processes, among others, are tightly regulated by over 560 gene proteases and 150 protease inhibitors encoded in the human genome (Lai et al, 2015), underscoring the vast diversity of peptides that can be created. Peptides play an important role in neurodegenerative diseases as exemplified by the amyloid-beta (Aβ) peptides in Alzheimer's disease (AD), which are generated by enzymatic cleavage of the amyloid precursor protein and are used in the clinic to support AD diagnosis (Blennow et al, 2006).

[1]German Center for Neurodegenerative Diseases (DZNE e.V.), Ulm 89081, Germany. [2]Department of Neurology, Ulm University Hospital, Ulm 89081, Germany. [3]Department of Neurology, Martin-Luther-University Halle-Wittenberg, Halle (Saale) 06120, Germany. ✉E-mail: patrick.oeckl@uni-ulm.de

Peptidomics is the large-scale analysis of peptides and aims at the identification and quantification of all peptides (peptidome) present in a sample (Foreman et al, 2021; Muqaku and Oeckl, 2022). In contrast to proteomics analysis where proteins are digested with proteases into peptides, the peptidomics approach involves separating peptides from proteins and analyzing them without enzymatic treatment. This enables the assessment of peptides in an intact form as they are present in the sample and improves sensitivity by reducing the concentration dynamic range of analytes. Many studies have focused on measuring peptides on a global scale in CSF of AD patients, but peptidomics data in other neurodegenerative diseases, including ALS, are sparse (Muqaku and Oeckl, 2022).

The aim of the present study was to identify and validate new biomarker candidates for ALS by peptidomic analysis of CSF samples. We used state-of-the-art mass spectrometry (MS) and an optimized peptidomics workflow for CSF to screen for peptide biomarkers in a discovery cohort with 24 ALS and 24 control (Con) patients. Selected candidate peptides were evaluated in a targeted approach with a validation cohort of 67 ALS and 42 Con CSF samples and correlated to clinical parameters.

# Results

## Analytical evaluation and characterization of the screening peptidomics workflow

We used a pooled CSF sample to evaluate and characterize the peptidomics workflow (Fig. 1A). By using 200 μL CSF, we could identify 10,710 peptides (Fig. 1B). Fractionation did not increase the number of peptide IDs in the CSF pool sample (Fig. 1B). Increasing the sample volume to 600 μL and applying fractionation further increased the number of peptide IDs to 16,303 (Fig. 1B). Since a volume of 600 μL CSF is often not available for biomarker studies, we considered 200 μL CSF as appropriate and this volume was used in all experiments. The number of peptide IDs did not change continuously after up to five freeze–thaw cycles (Fig. 1C) and after overnight sample incubation at room temperature (RT, Fig. 1D). In terms of the number of identified peptides, the method showed a good reproducibility with coefficient of variation (CV) of 2.7% ($n = 5$, Fig. 1E).

The quantitative analysis of five replicate CSF pool samples revealed a CV < 35% for the five non-human standard peptides (StdPep) when the data were either not normalized or normalized to the total ion chromatogram (TIC) and a CV < 20% for the normalization to StdPep (Fig. 1F). Looking at all peptides identified in the CSF pool sample, a median CV < 20% was obtained for peptides quantified in all five replicates independent of the normalization procedure (No, TIC and StdPep, Fig. 1G). Based on these results and because the approaches without and with TIC normalization in the QC samples might underestimate the variation in patient samples, we decided to use the normalization to the StdPep in subsequent experiments especially because it is independent from variations from highly abundant peptides. Peptide quantities normalized to StdPep for all quantified peptides correlated very strongly between five replicates with a Spearman's correlation coefficient over 0.94 (Fig. 1H). The dynamic range of

peptide abundance quantified in five replicates reached almost seven orders of magnitude (Fig. 1I).

## Screening peptidomic analysis in the discovery cohort

We performed a peptidomic analysis in CSF samples from a discovery cohort of patients consisting of 24 Con and 24 ALS patients (Table 1). Three quality control CSF pool samples (QC) were included to monitor analytical performance. In the patient cohort we could identify a total of 33,605 peptides (Dataset EV1). The number of identified peptides over all patients showed low variation (CV = 6.1%, $n = 48$, Fig. 2A). A median CV < 22% was obtained for peptides quantified in all QCs with small differences between three normalization approaches (Fig. 2B). The normalization to StdPep showed the best results with respect to the variation of StdPep in QCs (CV < 20%, $n = 3$, Fig. 2C) and patient samples (CV < 40%, $n = 48$, Fig. 2D). The dynamic range of peptide abundances in samples spanned almost eight orders of magnitude (Fig. 2E).

The group comparison analysis revealed 56 peptides differentially regulated ($P$ value < 0.05, s0 = 0.1) in ALS compared to control samples (Fig. 2F; Dataset EV2). All of them showed a change of more than 100% which exceeded the observed analytical variation in patient samples. The hierarchical clustering analysis performed with the significantly changed peptides clearly separated the two patient groups with only one control sample grouped into ALS (specificity of 96%, Fig. 2G). Also, the principal component analysis (PCA) with the significantly changed peptides distinguished the two patient groups with very small overlap (Fig. 2H).

A search of the peptidomics data against a peptide library created from the peptidatlas FASTA file from peptipedia 2.0 (Cabas-Mora et al, 2024) identified only 217 additional peptides with unaltered abundance in ALS compared to Con.

The pathways for generating active peptides from precursor proteins also comprises proteases which cleave C-terminal of arginine (R) and lysine (K) residues and releasing peptides preceded by R/K in the precursor protein sequence and/or ending with R/K. These are also the specific cleavage sides of trypsin widely used in proteomic analysis for protein digestion which is why we called them tryptic-like peptides. A higher number of tryptic-like peptides might indicate a disease-driven increase in protease activity. Indeed, the number of tryptic-like peptides (Fig. 2I; Appendix Table S1) and the % of tryptic-like peptides (Fig. 2J; Appendix Table S1) were significantly increased in ALS.

## Selection of the best peptide candidates for further validation

To further reduce the number of peptide candidates and select the most promising ones for the validation cohort, we developed a label-free targeted PRM method by using CSF pool samples instead of stable isotope-labeled standard peptide (SISPep) (Fig. 3A). The final label-free PRM method, consisting of 41 peptides, was applied to 10 Con and 12 ALS CSF samples from the discovery cohort. The statistical analysis of PRM data revealed 11 peptides significantly regulated in ALS compared to controls (Fig. 3B; Appendix Table S2), and the regulation direction, up- and down, matched with the results from the screening experiment.

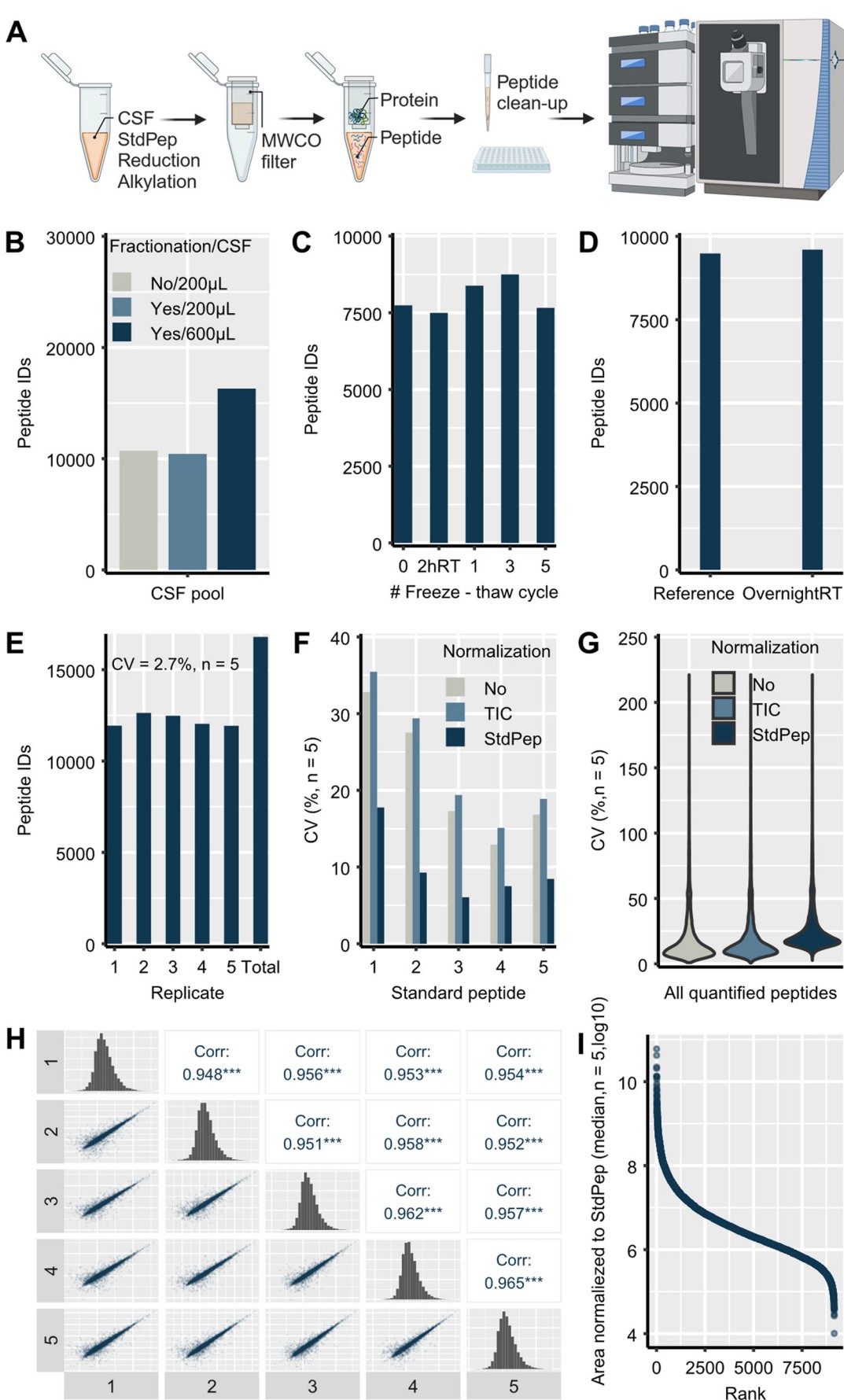

**Figure 1. Analytical evaluation of the screening peptidomics method.**

The peptidomics workflow starts by adding the standard peptides to the samples and reduction/alkylation of protein/peptides, continues with separation of peptides from proteins using molecular weight cut off filter (MWCO), purification of isolated peptides and finally mass spectrometry analysis (**A**). The number of peptides identified in a pooled CSF sample when using 200 μL with- and without sample fractionation and 600 μL followed by fractionation of the isolated peptides (**B**). The number of peptide IDs after several freeze–thaw cycles (**C**) and overnight incubation at room temperature (**D**) of the sample. The reference sample has 0 freeze–thaw cycle and 2hRT indicates incubation for 2 h at room temperature. The number of peptide IDs in each of five technical replicates (1–5) and in all of them (Total, **E**). Coefficient of variation (CV) over five replicates of non-human standard peptides (StdPep, **F**) and of peptides quantified in all replicates (**G**). No—data were not normalized, TIC—normalization to total ion chromatogram, StdPep— normalization to StdPep. Multi scatterplot show the correlation and Spearman's correlation coefficient of quantified peptides over five replicates (**H**). ***$P$ value < 0.001. The abundance dynamic range of peptides quantified in all replicates is shown by ranking the peptides according to their abundance (**I**). Source data are available online for this figure.

**Table 1. Demographic characteristics of the CSF discovery, validation and neurodegeneration cohort.**

| | N (f/m) | Age (year)[a] | Mutations | ALSFRS-r[a] | Disease duration at LP (month)[a] |
|---|---|---|---|---|---|
| **Discovery cohort** | | | | | |
| Con | 24 (12/12) | 67 (57–71) | – | – | – |
| ALS | 24 (12/12) | 67 (59–72) | – | 42 (37– 43) | 12 (7.8–20.2) |
| $P$ value[b] | 1 | 0.975 | | | |
| **Validation cohort** | | | | | |
| Con | 42 (22/20) | 58 (49– 64) | – | – | – |
| ALS | 67 (31/36) | 58 (51– 64) | – | 43 (39– 46) | 11.4 (7–14.8) |
| sALS | 44 (23/21) | 59 (51– 67) | – | 43 (39– 46) | 11.7 (7.5–15.5) |
| gALS | 23 (8/15) | 57 (52– 63) | 12 × C9orf72 (6/6) 11 × SOD1 (2/9) | 40 (36– 44) | 11 (7–12.8) |
| $P$ value[b] | 0.671 | 0.643 | – | 0.148* | 0.436* |
| $P$ value[c] | 0.327 | 0.717 | – | – | – |
| **Neurodegeneration cohort** | | | | | |
| Con | 17 (9/8) | 66 (57– 73) | | | |
| AD | 20 (14/6) | 72 (67– 75) | | | |
| bvFTD | 16 (7/9) | 65 (60– 67) | | | |
| PD | 15 (6/9) | 65 (54–72) | | | |
| $P$ value[c] | 0.2739 | 0.046[d] | | | |

*Con* controls, *ALS* amyotrophic lateral sclerosis, *sALS* sporadic ALS, *gALS* genetic ALS, *AD* Alzheimer´s disease, *bvFTD* behavioral variant frontotemporal dementia, *PD* Parkinson´s disease, *f* female, *m* male, *ALSFRS-r* ALS functional rating scale-revised, *LP* lumbar puncture
[a]Median and interquartile range.
[b]Two groups (Con, ALS): Chi-squared test for sex and Wilcoxon test. *sALS vs. gALS.
[c]More than two groups: Chi-squared test for sex and Kruskal–Wallis test and Dunn´s post hoc test.
[d]$P$ value > 0.05 for all comparisons between two groups.

A network MCL clustering analysis performed on the STRING database with the parent proteins of peptides included in the list for the development of the label-free PRM method identified two big clusters (Fig. 3C). The first cluster, consisting of proteins APOC1, APOC3, APOE, CO3, CO4B, KNG1, VTDB, and B2MG, which primarily are expressed in plasma cells, liver and skeletal system, is referred to as the metabolic cluster because of involvement of its proteins in metabolic processes. The second cluster, consisting of brain and neuronal proteins such as NFL, NFM, NFH, MAP1B, MOG, and MYP0, is termed the neurodegeneration cluster due to the implication of its proteins in neurodegenerative diseases.

## Evaluation of the best peptide candidates in a validation cohort by targeted PRM

We developed a targeted PRM method using SISPep (Table EV1) to evaluate the most promising peptides in a larger validation cohort of 67 ALS and 42 Con patients (Table 1). From 11 peptides found regulated with label-free PRM method two NFM peptides, the CO3 peptide were excluded, and a second MAP1B peptide was added to the list for further evaluation. The APOC1_TP peptide did not fulfill the validation criteria and therefore was removed. All eight peptides included in the final PRM method were stable in CSF for at least five freeze–thaw cycles, showed good dilution linearity (deviation <20%) and a CV < 13% (Table 2). The PRM method with SISPep confirmed the correct peak picking in the label-free PRM experiments for NFL, APOC1_SE, CADM3, and SCG1 peptides. However, the MYL1 SISPep eluted more than one minute later than the endogenous peptide and thus did not confirm its identity (Fig. EV1).

In the screening peptidomics analysis, the MYL1 peptide was identified to be N-terminally acetylated in the first amino acid residue (A). Since acetylation has a mass shift of 42.01 Da, we

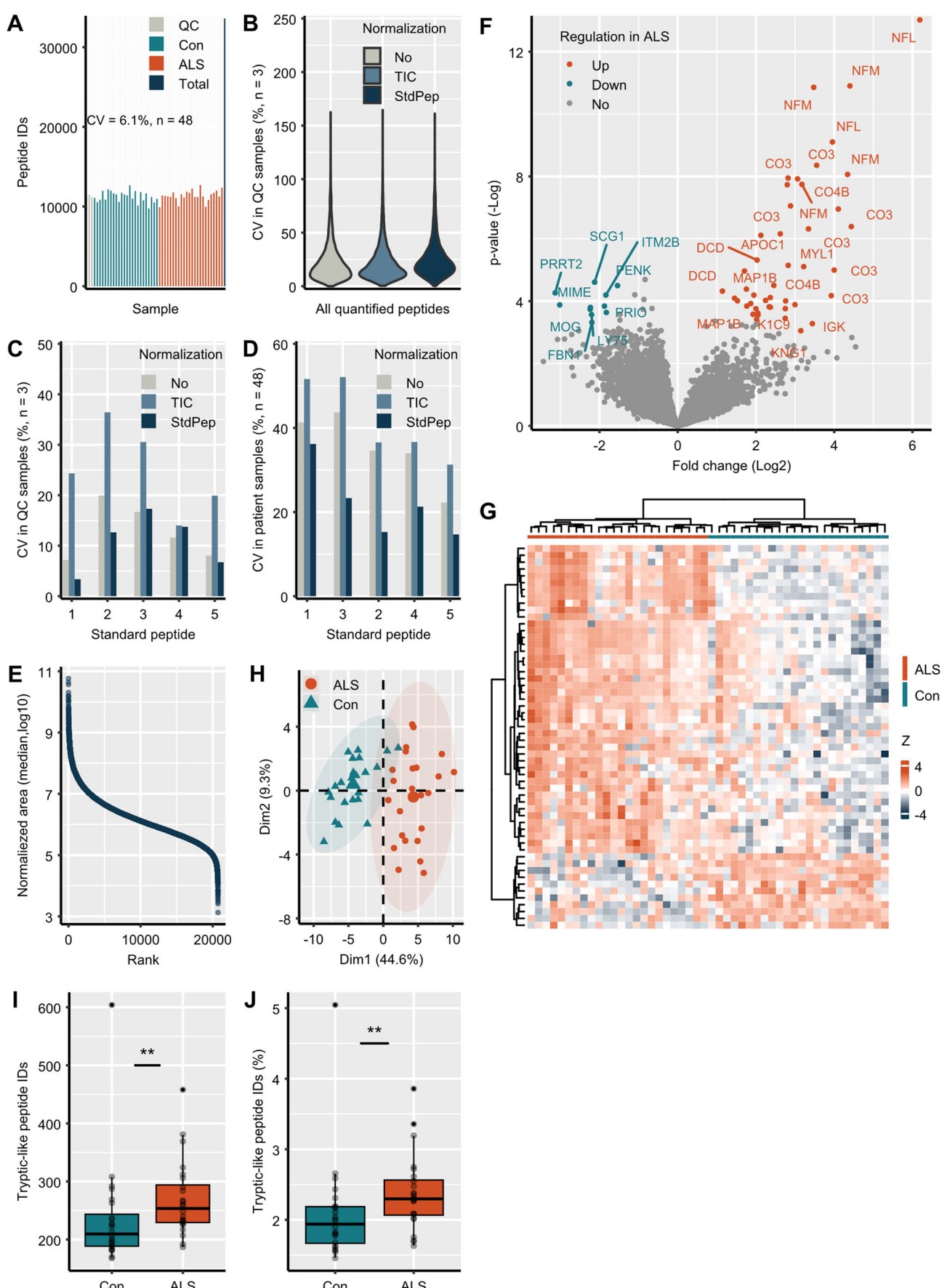

◀ **Figure 2.   Screening peptidomic analysis of ALS and control samples from the discovery cohort.**

The number of peptides identified in quality control samples (QC, $n = 3$), controls (Con, $n = 24$), amyotrophic lateral sclerosis (ALS, $n = 24$), and in total (all patients, $n = 48$, **A**). The variation in QCs of endogenous peptides (**B**) and non-human standard peptides (StdPep) (**C**) without data normalization (No), normalization to total ion chromatogram (TIC) and to StdPep. The variation of StdPep in patient samples (**D**). Abundance dynamic range of peptides in patient samples when normalizing the data to StdPep (**E**, $n = 48$). Volcano plot showing peptides significantly upregulated (orange), downregulated (green) and not regulated (gray) in CSF of ALS patients ($n = 24$) compared to non-neurodegenerative controls (Con, $n = 24$) based on Student´s *t* test and permutation-based FDR (**F**). Peptides are labeled with the Uniprot Entry Names of the proteins from which they derive. Due to space limitations, not all regulated peptides of a protein are labeled, but each protein is represented. Full list of peptides and significance is given in Dataset EV2. (**G**) Heatmap showing the results of hierarchical clustering analysis performed with significantly regulated peptides. Principal component analysis (PCA) performed with significantly regulated peptides (**H**). The ellipse was drawn with 95% confidence interval. Boxplots (minima, maxima, center, bounds of box and whiskers, and percentile) and single values of the number (**I**) and % (**J**) of tryptic-like peptides in Con ($n = 24$) and ALS ($n = 24$) samples. Tryptic-like peptides are peptides preceded by R/K in the precursor protein sequence and ending with R/K. **Wilcoxon test *P* value < 0.01.

searched a CSF pool sample separately with other modifications with the same mass shift including acetylation (K, $+42.01$ Da), trimethylation (KR and A, $+42.05$ Da) and guanidination (K, $+42.02$ Da). The highest identification score was obtained for trimethylation on K8 which was almost twice as high as for other modifications. Another search including all modifications (N-terminal acetylation, acetylation, trimethylation, and guanidination) identified the MYL1 peptide being trimethylated in the eighth amino acid residue (K). Finally, the SISPep confirmed the identity of the trimethylated MYL1 peptide (Fig. EV1).

PRM data were manually inspected and thereby no peaks were detected due to low abundance of endogenous peptide in seven samples for MYL1 and two samples for MAP1B_KE. The PRM analysis in the validation cohort showed good analytical performance based on the included QC samples (CV < 15%, $n = 7$). NFL, MYL1, APOC1 and MAP1B peptides were significantly upregulated in ALS, while peptides derived from CADM3, SCG1 and PENK were downregulated (Fig. 4A; Table 3; Appendix Table S3). These changes were similar in sALS and gALS with exception of PENK being significantly downregulated only in gALS (Fig. 4B). When stratified according to the mutation, PENK was especially downregulated in C9orf72 mutation carriers even compared to sALS (Fig. 4C). The abundance of CADM3 and SCG1 peptides was significantly reduced in C9orf72 mutation carriers compared to controls (Fig. 4C). The abundance of MYL1 and APOC1 peptides were significantly increased in male compared to female patients (Fig. EV2).

The receiver operating characteristic (ROC) curve analysis showed NFL with the best area under the curve (AUC) value of 94% and the AUC of the other peptides ranged from 61.2% to 84.2% (Fig. 4D). However, a combination of all peptides in a logistic regression model performed best with an AUC of 98% (Fig. 4D).

NFL is as a well-established biomarker for ALS, and its increase in ALS demonstrated the reliability of the peptidomic analysis. We compared our results with an established antibody-based assay (Ella) commonly used for NFL measurement. The PRM results for NFL correlated very well with Ella data with a Spearman's correlation coefficient ($r$) of 0.968 (Fig. 4E). The ROC analysis with Ella data for NFL (AUC = 96.4%) showed a slightly lower discrimination performance than the all peptide approach (Fig. 4D).

We observed significant correlations of peptide levels within the group of peptides upregulated in ALS and within the group of downregulated peptides (Fig. 5; Appendix Table S4). Particularly, the abundance levels of the two MAP1B peptides correlated strongly with each other ($r = 0.87$) and with the NFL peptide

($r = 0.82$ and $r = 0.84$). A negative correlation with the FRS_r (ALS functional rating scale-revised) score was observed for NFL ($r = -0.40$) and MAP1B peptides ($r = -0.32$ and $r = -0.42$). Only the NFL peptide correlated with disease duration ($r = -0.25$). A significant correlation with age in all patients of the validation cohort was observed for the MAP1B_EA peptide ($r = 0.325$) and the NFL peptide ($r = 0.24$). In Con patients, peptides of MAP1B ($r = 0.41$ and $r = 0.55$), NFL ($r = 0.81$), and MYL1 ($r = 0.35$) correlated with age (Fig. EV3). Only the group of upregulated peptides showed significant correlations with the albumin quotient (Qalb) to various degrees ($r$: MYL1 $= 0.41$, MAP1B $= 0.56$ and $0.58$, NFL $= 0.68$, APOC1 $= 0.83$ (Fig. EV3).

## Evaluation of peptide candidates in other neurodegenerative diseases

To evaluate the specificity for ALS of the eight peptide candidates, we investigated their levels with the developed PRM in CSF of other neurodegenerative diseases, including 20 patients with Alzheimer´s disease (AD), 16 behavioral variant frontotemporal dementia (bvFTD), 15 Parkinson´s disease (PD) and 17 non-neurodegenerative control patients (Con, Table 1).

PRM data were manually inspected and no or very low signal was detected in 14 samples for MAP1B_EA, 15 samples for MYL1, 19 samples for APOC1 and 6 samples for PENK. The MAP1B_KE peptide was excluded from further evaluation due to the absence of signal in a large number of samples. Additionally, a transition of MAP1_EA and PENK showed matrix interferences in several samples and was not included in the ratio calculation for both peptides. The PRM analysis showed good analytical performance based on the included QC samples (CV < 15%, $n = 5$).

The results showed significant changes only for NFL, whereas the other peptides were not changed (Fig. 6; Appendix Table S5). The ratios of the MYL1 peptide to the acetylated and methylated SISPep correlated strongly with each other ($r = 0.992$, Fig. EV4).

## Discussion

In the present study, we identified and validated eight CSF peptides as novel biomarker candidates for ALS. In total, we detected 33,605 peptides in CSF samples from a discovery cohort of ALS and control patients by peptidomics, 56 of them were significantly regulated in ALS compared to controls. A systematic selection for the best candidates led to a targeted PRM method with eight peptides (including peptides from NFL, MAP1B, MYL1, APOC1,

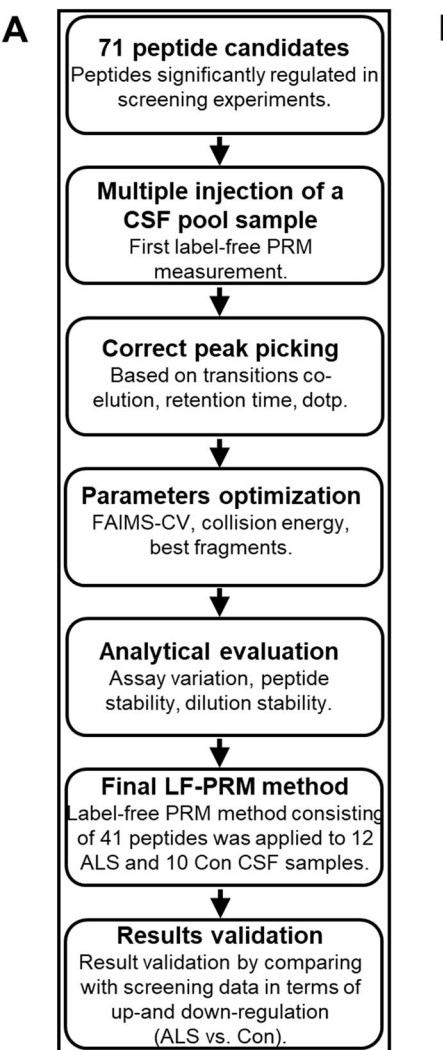

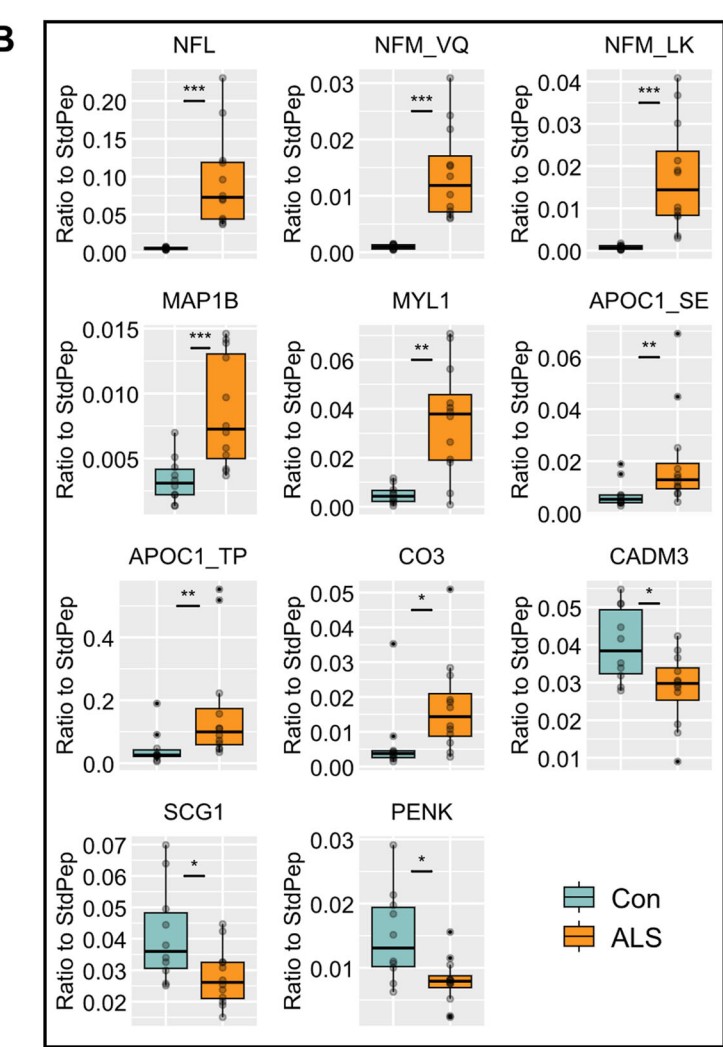

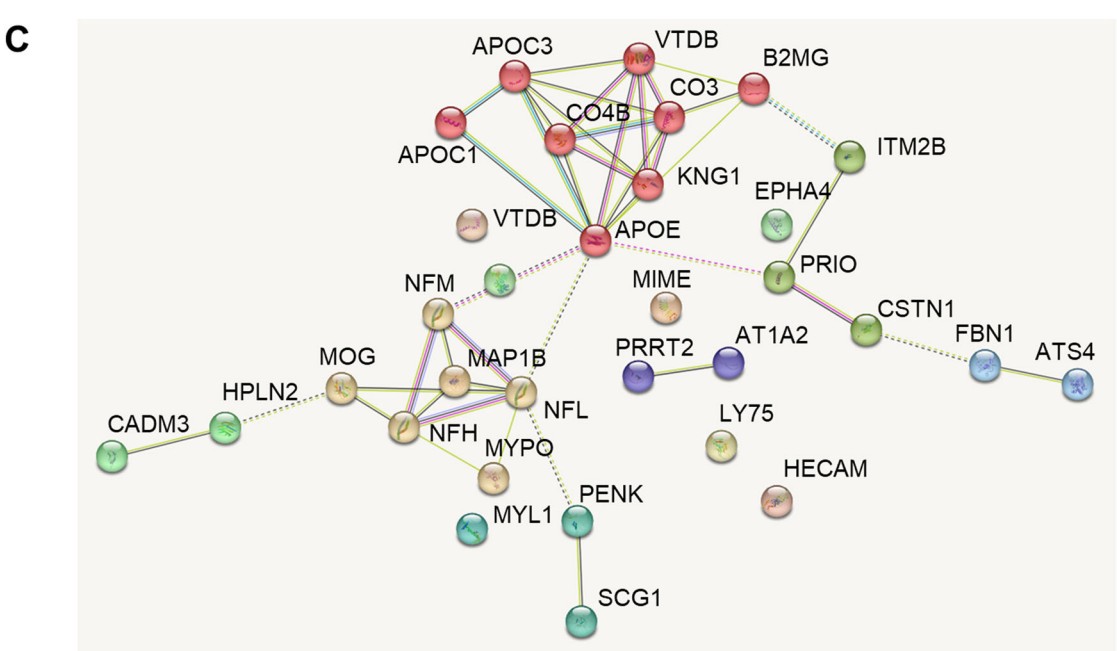

**Figure 3. Label-free targeted PRM analysis of samples from the discovery cohort.**

Steps followed during the development of the label-free parallel reaction monitoring (PRM) method by using a CSF pool sample instead of synthetic standard peptides (A). Boxplots (minima, maxima, center, bounds of box and whiskers, and percentile) with individual values show the data of label-free PRM analysis for peptides found significantly regulated by comparing 12 ALS and 10 Con samples from the discovery cohort (B). Peptides derived from the same proteins are distinguished by writing the sequence of the first two amino acids after the protein Uniprot Entry Name. Data were normalized to five non-human standard peptides (StdPep) added to the sample and used as global standards. The Wilcoxon test was applied for group comparison analysis. *P value < 0.05; **P value < 0.01; ***P value < 0.001. Result of network MCL clustering analysis performed in the STRING database with parent proteins of peptides found significantly regulated in discovery experiments (C). Source data are available online for this figure.

**Table 2. PRM assay performance.**

| Protein/peptide | Stability test (%, n = 2) | | | | Dilution stability (%, n = 2) | | | Intra-assay variation (%, n = 5) |
|---|---|---|---|---|---|---|---|---|
| | 2 h RT | 1 cycle | 3 cycles | 5 cycles | 1–2 | 1–4 | 1–8 | |
| APOC1 | 107–98 | 101– 94 | 117– 123 | 148– 123 | 100– 113 | | | 5.2 |
| CADM3 | 90– 110 | 96– 101 | 111– 99 | 100– 106 | 119– 119 | 112– 111 | 104– 116 | 12.2 |
| MYL1 | 99– 106 | 89– 105 | 94– 110 | 90– 101 | 96– 99 | 85– 80 | | 4.6 |
| NFL | 98– 92 | 117– 92 | 91– 89 | 104– 91 | 98– 94 | | | 10.7 |
| SCG1 | 100– 99 | 97– 89 | 89– 99 | 104– 103 | 92– 104 | 99– 116 | 99– 112 | 4.6 |
| MAP1B_KE | 111– 97 | 105– 106 | 108– 111 | 107– 112 | 109– 106* | 89– 88* | | 8.6 |
| MAP1B_EA | 60– 102 | 105– 104 | 101– 102 | 97– 104 | 89– 83 | 83– 97* | | 5.7 |
| PENK | 116– 95 | 107– 95 | 114– 104 | 108– 92 | 103– 114 | | | 5.3 |

2 h RT—incubated for 2 h at room temperature; 1 cycle—one freeze–thaw cycle; 3 cycles—three freeze–thaw cycles; 5 cycles—five freeze–thaw cycles.
*Evaluated with a shorter LC gradient.

CADM3, SCG1, and PENK). Evaluation of the eight most promising peptides in a validation cohort confirmed findings from the label-free experiment and uncovered differences between sALS and gALS. The parent proteins of the peptides cover key processes such as neurodegeneration, metabolic alteration, muscle atrophy/wasting, synaptic loss, and/or deregulation in vesicle transport or the secretory pathway. The abundance of the NFL peptide in CSF showed a very strong correlation with an established immunoassay for the NFL protein. ROC curve analysis performed with a combination of the eight peptides revealed a better diagnostic performance than single peptides (including NFL). In addition, only the NFL was found to be deregulated in other tested neurodegenerative diseases, suggesting that the peptide biomarker candidates are specific to ALS.

Although the parent proteins of several peptide candidates have been found deregulated in neurodegeneration (Barschke et al, 2022; Holtta et al, 2015; Morgan and Carlyle, 2024; Quinn et al, 2023), all peptide sequences are reported here for the first time as being deregulated in neurodegenerative diseases, including NFL. MAP1B illustrates the relevance of targeting specific peptide sequences, as only two of the ten quantified peptides showed increased levels in ALS. Considering the average abundance of peptides, as is commonly done in proteomics, would have failed to reveal the changes in MAP1B protein. Indeed, proteomics identified that the MAP1B protein was not increased in ALS (Oeckl et al, 2020). To the best of our knowledge, we detected MYL1 for the first time in the CSF of patients with neurodegenerative diseases, and together with MAP1B and PENK, it has not previously been linked to ALS.

Cluster analysis with parent proteins of peptides significantly regulated in screening experiments revealed a metabolic and neurodegenerative cluster. Both clusters are represented in the

final targeted PRM method by APOC1, NFL and MAP1B peptides. Other four remaining peptides of the PRM method derived from MYL1, CADM3, SCG1, and PENK. MYL1, the non-regulatory myosin light chain protein that builds skeletal muscles, is likely regulated in response to muscle atrophy and wasting in ALS patients (Ravenscroft et al, 2018; Rayment et al, 1993; Reggiani et al, 2000). The other three proteins, which are synaptic and/or vesicle-located, are downregulated in ALS, possibly reflecting neuronal loss, synaptic dysfunction and/or dysregulation of vesicle transport and the regulated secretory pathway (Schrott-Fischer et al, 2009; Sjostedt et al, 2020).

The significantly increased abundance of peptides from the neurofilament proteins (NFL, NFH and NFM) in the discovery peptidomics experiment supports the reliability of our peptidomics analysis. Higher CSF levels of all three neurofilament proteins have been reported previously (Oeckl et al, 2020), and NFL is a well-established marker for neurodegeneration and the most prominent ALS biomarker to date (Steinacker et al, 2016). Since most NFL in CSF is fragmented (Budelier et al, 2022), the higher levels of NFL peptides in our study are a proof-of-principle. As expected the NFL showed elevated abundance level also in ALS samples from the validation cohort.

We observed upregulation of two MAP1B peptides in ALS. MAP1B is a brain-enriched protein but is also expressed in many other organs (Halpain and Dehmelt, 2006; Sjostedt et al, 2020; Tucker and Matus, 1987). MAP1B is highly expressed during early neuronal development and thereby is involved in the formation and development of axons and dendrites through regulation of stability and dynamic turnover of microtubules (Halpain and Dehmelt, 2006; Tortosa et al, 2011). In adulthood, MAP1B is expressed in brain areas showing high degree of plasticity, such as the olfactory

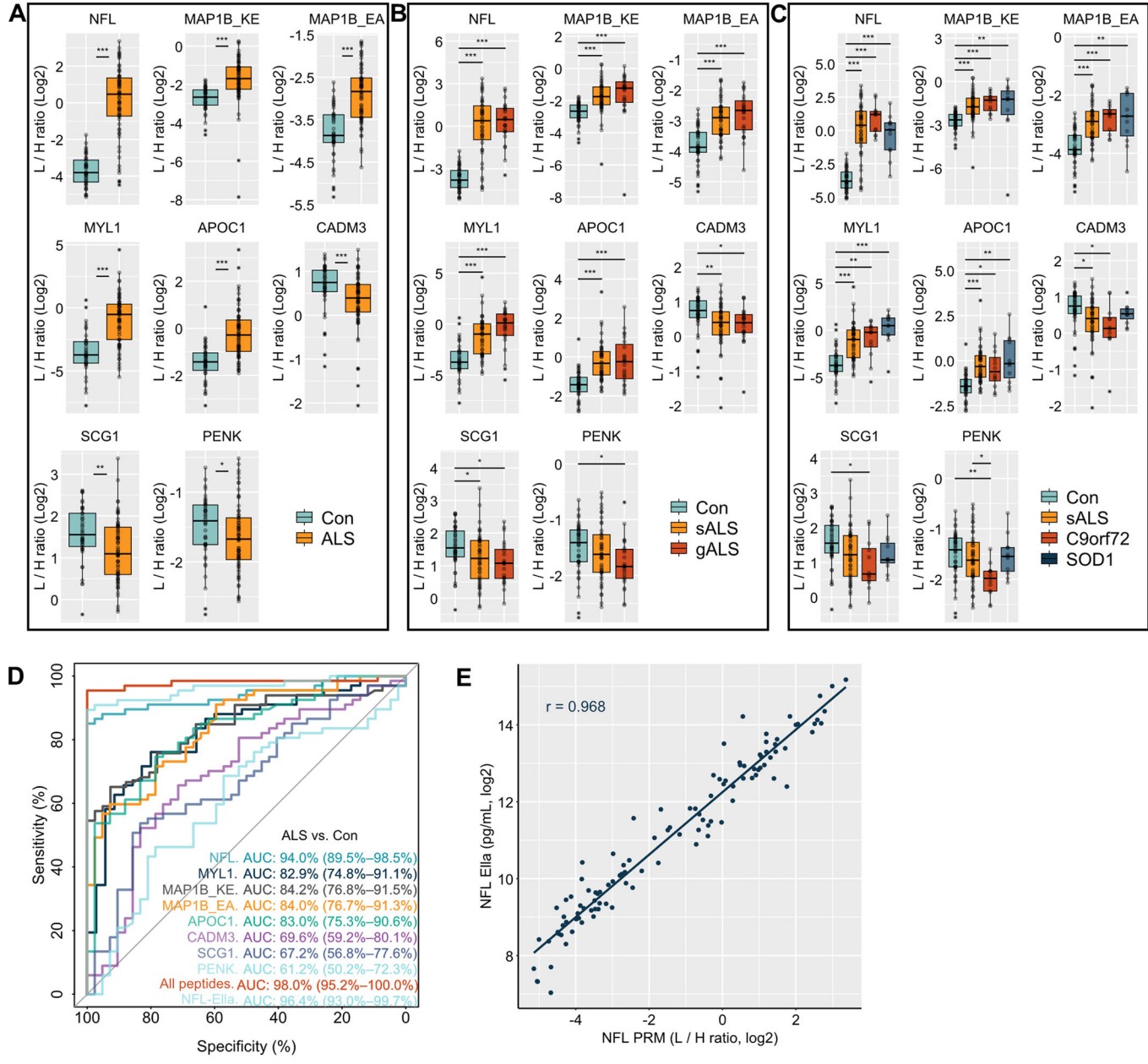

**Figure 4. Targeted PRM analysis of peptide candidates in the validation cohort.**

Eight peptides selected during the discovery experiments were measured by targeted PRM in cerebrospinal fluid (CSF) of amyotrophic lateral sclerosis (ALS, n = 67) patients and non-neurodegenerative controls (Con, n = 42, **A**). ALS patients were divided into sporadic (sALS, n = 44) and genetic ALS (gALS, n = 23, **B**) patients and further into *C9orf72* (n = 12) and *SOD1* mutation carriers (n = 11, **C**). We had 7 missing values for MYL1 peptide in Con samples and 2 for MAP1B_KE peptide (1 Con and 1 sALS). For each peptide the data were normalized to their respective stable isotope-labeled standard peptide (SISPep) indicated as light to heavy ration (L/H ratio). The two different MAP1B peptides are indicated by the first two amino acids of the peptide sequence. Data are visualized by boxplots (minima, maxima, center, bounds of box and whiskers, and percentile) and individual values. Wilcoxon test was applied for comparison of two groups and the Kruskal–Wallis test corrected with Dunn´s post hoc test for multiple comparison for >2 groups. Receiver operating characteristic (ROC) curve analysis of each peptide and the combination of all peptides (**D**). Numbers in brackets after the area under the curve (AUC) value represent 95% confidence interval. Scatterplot and a linear regression line of PRM and ELLA data for NFL in CSF samples of the validation cohort (**E**). Correlation analysis was performed by Spearman's correlation coefficient. *P value < 0.05; **P value < 0.01; ***P value < 0.001. Source data are available online for this figure.

bulb, olfactory epithelium, and the hippocampus (Villarroel-Campos and Gonzalez-Billault, 2014). It is mainly present in dendrites and was identified in mouse synapses in proteomics experiments (Collins et al, 2005; Tortosa et al, 2011). However, despite MAP1B´s role in cellular structural stability (Oberstadt et al, 2018), MAP1B has been attributed regulatory function in morphology and physiology of synapses (Bodaleo et al, 2016; Tortosa et al, 2011) and is involved in the pathomechanisms of spinal muscular atrophy (Bora et al, 2021). Others reported the regulation of MAP1B expression and its homologs by proteins

**Table 3. Peptides significantly regulated in the validation cohort.**

| Protein Acc. Nr. | Uniprot entry name | Gene name | Protein name | Peptide sequence | Sequence Start-End | Length (#AA) | Regulation in ALS |
|---|---|---|---|---|---|---|---|
| P07196 | NFL | NEFL | Neurofilament light polypeptide | SFPSYYTSHVQEEQIEVEETIEAAKAEEA | 438 – 466 | 29 | Up |
| P46821 | MAP1B | MAP1B | Microtubule-associated protein 1B | KEPKKEIKKLPKDAKKSSTPLS | 726 – 747 | 22 | Up |
| P46821 | MAP1B | MAP1B | | EAKKPAALKPKVPKKEESVKKDSVA | 748 – 772 | 25 | Up |
| P05976 | MYL1 | MYL1 | Myosin light chain 1/3, skeletal muscle isoform | APKKDVKK(trimethylation)PVAAAAAAPAPAPAPAPAPAPAPAKPKEE | 2 – 36 | 35 | Up |
| P02654 | APOC1 | APOC1 | Apolipoprotein C-I | SETFQKVKEKLKIDS | 69 – 83 | 15 | Up |
| Q8N126 | CADM3 | CADM3 | Cell adhesion molecule 3 | TLNVNDPSPVPSSSSTY | 313 – 329 | 17 | Down |
| P05060 | SCG1 | CHGB | Secretogranin-1 | EPRAYFMSDTREE | 425 – 437 | 13 | Down |
| P01210 | PENK | PENK | Proenkephalin-A | DGTSTLRENSKPEESHLL | 79 – 96 | 18 | Down |

known to be implicated in ALS pathology, such as TDP43 and FUS (Coyne et al, 2014; Garone et al, 2023; Majumder et al, 2016; Oberstadt et al, 2018; Romano et al, 2016; Strohm et al, 2022), thus linking MAP1B directly with ALS. In our data, the abundance of MAP1B peptides show a strong correlation with NFL and therefore their upregulation might be linked with neurodegeneration. Importantly, the two peptides targeted in our study are localized in the microtubule binding domain of MAP1B heavy chain (Noble et al, 1989; Togel et al, 1998). It remains to be investigated whether the specific regulation of these two peptides is of functional relevance and to what extend their upregulation reflect perturbation in the function of dendrites and synapses or similar to NFL were released upon neuronal degeneration.

CSF levels of trimethylated MYL1 peptide were increased in ALS. MYL1 is part of the myosin machinery participating in muscle function. The measured MYL1 peptide in this study is located N-terminally and more precisely is part of the MYL1 N-terminal extension. The function of the MYL1 N-terminal extension as shown in animal model is to maintain the integrity of myosin, modulate force generation and enhance myosin attachment to actin (Kazmierczak et al, 2009). Moreover, the MYL1 N-terminal extension undergo a transient interaction with motor domain of the myosin head during the ATPase cycle (Logvinova et al, 2018). Thus, the upregulation of MYL1 peptide in ALS might be caused by impaired motor processes in muscle and not only due to structural changes or muscle wasting. A further elevated MYL1 CSF level in gALS and SOD1 mutation carriers even compared to sALS is in line with reports of SOD1 mutation targeting primary skeletal muscles (Dobrowolny et al, 2008), additionally emphasizing the idea of muscle functional alteration impacting elevated MYL1 levels. How methylation could influence the MYL1 peptide levels needs additional studies, but the unmodified MYL1 peptide was not identified. In general, males tend to have a higher muscle mass than females, which may predispose them to higher level of MYL1 peptide. Indeed, the abundance of MYL1 peptide is significantly higher in males than females, supporting the notion of MYL1 changing in a systemic fashion.

APOC1 is a plasma protein and constituent of very low-density lipoprotein (VLDL) and high-density lipoproteins (HDL) (Rouland et al, 2022), but it is synthesized also in the brain (Abildayeva et al, 2008). In a proteomics study, we found upregulation of other apolipoproteins in CSF of ALS patients (Oeckl et al, 2020). The observed weight loss in ALS patients, accompanied by metabolic alterations, is linked to an increased energy demand that is compensated by lipid mobilization (Burg and Van Den Bosch, 2023; Guillot et al, 2021). It would be very speculative to specify the impact of elevated APOC1 CSF level when taking into account the contradictory evidence regarding the effects of lipids and lipoproteins in ALS (Burg and Van Den Bosch, 2023; Guillot et al, 2021; Ingre et al, 2020). However, the role of APOC1 in lipid metabolism and transport is suggestive for its upregulation in CSF samples of ALS patients. APOC1 levels in the data from the validation cohort differ significantly between female and male individuals, pointing to the well-known gender dependent differences in fat mass and might indicate a systemic rather than central nervous system (CNS) specific alteration of APOC1 levels.

PENK is a precursor protein of many neuropeptides acting as a group of endogenous opioids (Fricker et al, 2020). PENK is highly expressed in the brain but it has been found in other non-neuronal tissues, too (Denning et al, 2008; Sjostedt et al, 2020). It plays a role in processes related to pain perception and stress response (Przewlocki and Przewlocka, 2001). PENK is produced by the striatal medium spiny projection neurons (MSNs), therefore its reduced levels in Huntington's disease characterized by striatal atrophy and/or dysfunction has been linked with degeneration of MSNs (Barschke et al, 2022; Niemela et al, 2021). In agreement with our previous observation using targeted proteomics (Barschke et al, 2022), the PENK peptide was unchanged in CSF of sALS patients, but we here observed a specific downregulation in C9orf72. The specific patterns of thalamo-cortico-striatal atrophy (Nigri et al, 2023) and more severe striatal pathology (Cykowski et al, 2017) of C9orf72 mutation carriers versus non-carrier ALS patients could explain the specific PENK downregulation in c9ALS. Consistent with this, the staging efforts of Heiko Braak showed an involvement of the striatum in stage 3; the neuroanatomical alterations of patients carrying C9orf72 mutations were more severe than in the majority of patients (Brettschneider et al, 2013). Further functional studies are required to conclude whether downregulation of PENK in C9orf72 mutation carriers reflect a high degree of striatal dysfunction.

SCG1 is part of acidic and secretory proteins of the granin family (Bartolomucci et al, 2011; Helle, 2004; Kromer et al, 1998) and localized in large dense-core vesicles (LDCV) of endocrine and neuroendocrine systems as well as neurons, thus making it a good surrogate marker for LDCV and synaptic alteration (Eder et al, 1998; Marksteiner et al, 2000; Schrott-Fischer et al, 2009; Winkler and Fischer-Colbrie, 1998). Immunohistochemical staining

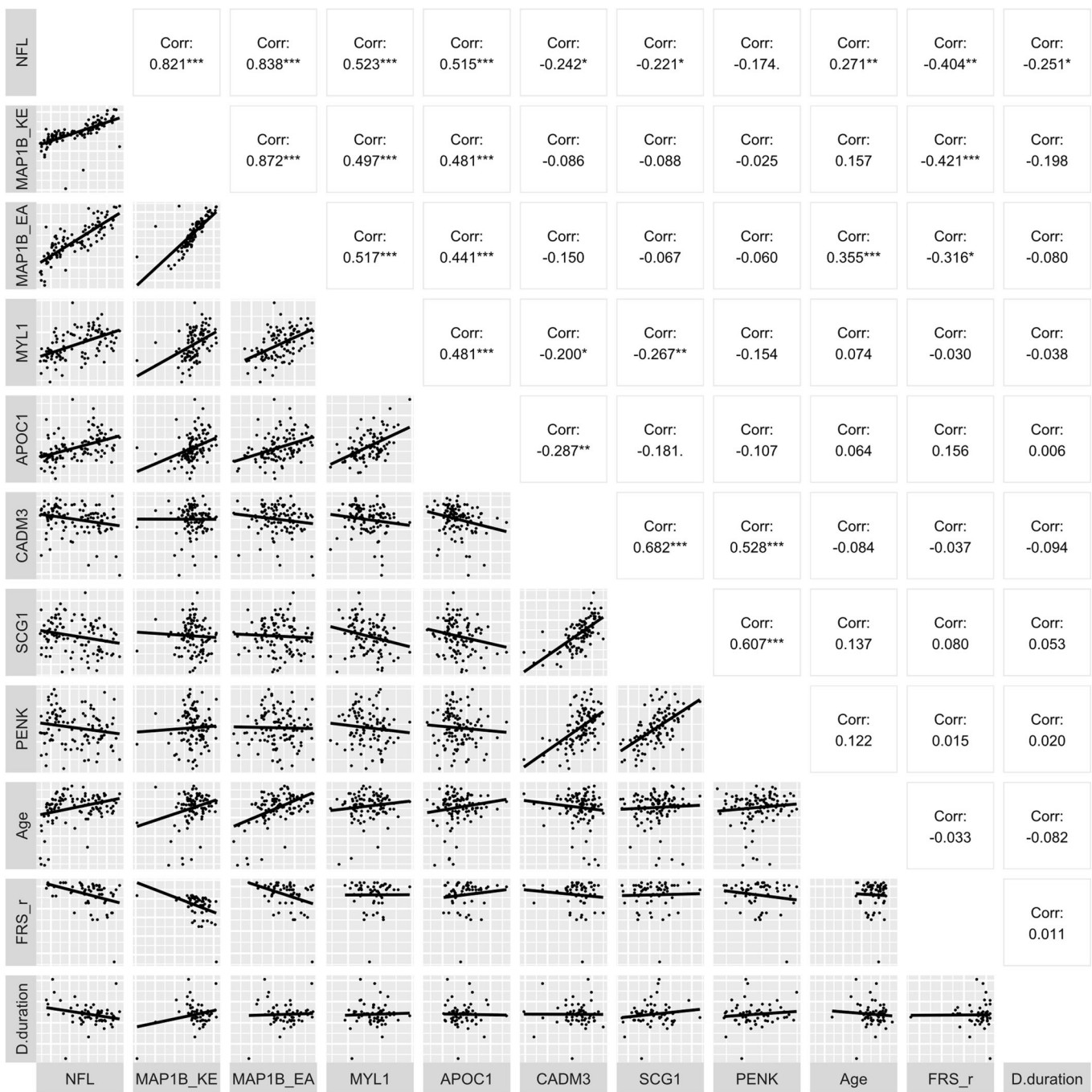

**Figure 5. Correlation of eight peptides and clinical data of patients in the validation cohort.**

Multi scatterplot show the correlation and Spearman's correlation coefficient for eight peptides and clinical data. All data were log2 transformed, and each dot represents the data of a patient. The two different MAP1B peptides are indicated by the first two amino acids of the peptide sequence. FRS_r - ALS functional rating scale-revised, D. duration – disease duration at lumbar puncture. *P value < 0.05; **P value < 0.01; ***P value < 0.001. Source data are available online for this figure.

revealed reduced intensity of SCG1 peptides in tissue samples of ALS patients due to a loss of neurons, while in the remaining neurons intracellular accumulation of SCG1 peptides in SOD1 aggregates was observed (Schrott-Fischer et al, 2009). Similar to our peptidomics results, lower CSF levels of SCG1 protein have been reported for ALS, Parkinson's disease and AD patients and were explained with neuronal loss or reduction in protein secretion

(Park et al, 2020; Zhu et al, 2019). The downregulation of CSF SCG1 peptide abundance in ALS patients may be attributed to perturbation of secretory pathways, neuronal loss, synaptic dysfunction, decreased density of SCG1 in the neuropil and intracellular accumulation in SOD1 aggregates (Schrott-Fischer et al, 2009). SCG1 specifically interacts with mutant forms of SOD1 (Urushitani et al, 2006). Although the P413L SCG1 variant has

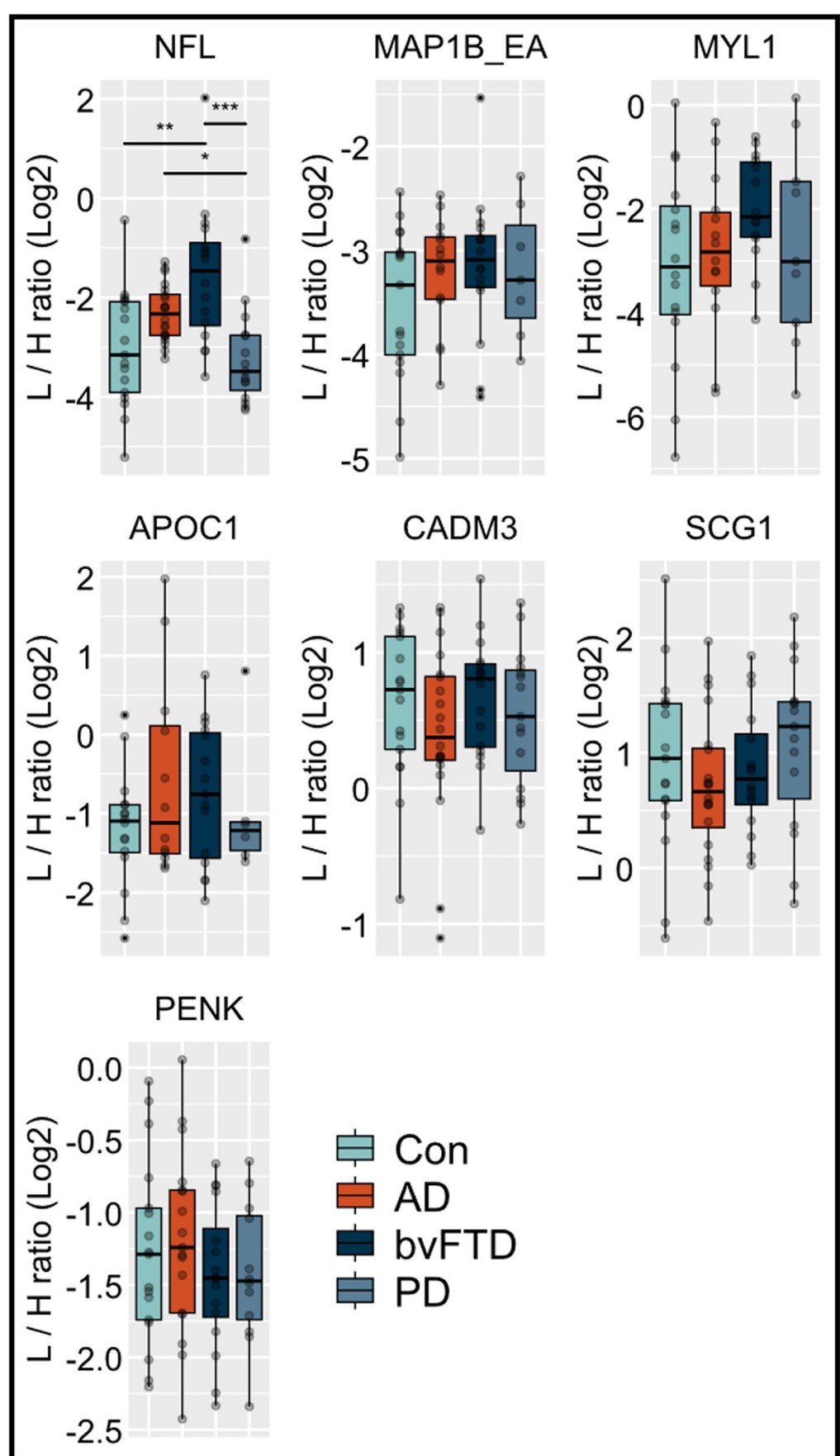

**Figure 6. Targeted PRM analysis of peptide candidates in the neurodegeneration cohort.**

Eight peptides were measured by targeted PRM in cerebrospinal fluid (CSF) of Alzheimer´s disease (AD, $n = 20$), behavioral variant frontotemporal dementia (bvFTD, $n = 16$), Parkinson´s disease (PD, $n = 15$) patients and non-neurodegenerative controls (Con, $n = 17$). We had missing values in 14 samples for MAP1B_EA (6 AD and 8 PD), 15 samples for MYL1(1 Con, 6 AD, 2 bvFTD and 6 PD), 19 samples for APOC1 (1 Con, 8 AD, 1 bvFTD and 9 PD) and 6 samples for PENK (3 AD and 3 PD). For each peptide, the data were normalized to their respective stable isotope-labeled standard peptide (SISPep) indicated as light to heavy ratio (L/H ratio). Data are visualized by boxplots (minima, maxima, center, bounds of box and whiskers, and percentile) and individual values. Group comparison analysis was performed with Kruskal–Wallis test corrected with Dunn´s post hoc test for multiple comparison. *$P$ value < 0.05; **$P$ value < 0.01; ***$P$ value < 0.001. Source data are available online for this figure.

been reported as a risk factor and modifier of disease onset for ALS (Gros-Louis et al, 2009), it was not confirmed in other populations (Ricci et al, 2015; van Vught et al, 2010). It remains to be determined in future studies whether reduced levels of SCG1 in ALS associated with disease stage and disease progression.

CADM3 is a synaptic and brain-enriched immunoglobulin-like cell–cell adhesion molecule which plays an important role in the formation of synapses, axon bundles and myelinated axons (Biederer et al, 2002; Gao et al, 2008; Kakunaga et al, 2005). A previous proteomic study reported downregulation of the CADM3 protein in ALS CSF samples (Collins et al, 2015). This agrees with our peptidomics data targeting a peptide located in the C-terminus of the extracellular part of CADM3. In peripheral neurons, the axon-glial interaction is mediated by CADM3. Recently, it has been reported that a CADM3 mutation inducing abnormal axon-glia interaction causes Charcot-Marie-Tooth disease which shares with ALS the impaired motor function (Rebelo et al, 2021). The intracellular retention and reduced cell surface expression of mutant CADM3 protein (Rebelo et al, 2021) could explain the reduction of CSF CADM3 peptide in ALS. Considering its synaptic and axonal localization, the deregulation of CADM3 might indicate alterations in synaptic function and impaired integrity of cell–cell communication in ALS patients.

In the regulated secretory pathway, precursor proteins are cleaved amongst others by proteases at C-terminal basic amino acids (K/R), thereby releasing tryptic-like peptides with sequences preceded by K/R and ended with K/R (Bergeron et al, 2000; Cawley et al, 2012). Removing of the basic amino acids by carboxypeptidase E (CPE) give raise to the active form of hormones and neuropeptides (Bergeron et al, 2000; Cawley et al, 2012). Here, we postulate that the significant increase of the number of tryptic-like peptides in ALS samples in the peptidomics data indicates changes in the secretory pathway, either due to increased enzymatic activity generating more tryptic-like peptides or malfunctioning of CPE leading to accumulation of unprocessed tryptic peptides.

We demonstrate the complexity of the CSF peptidome and the suitability of peptidome analysis for biomarker research. To date, the highest number of peptides identified in CSF of neurodegenerative disease patients is 18,031, obtained through extensive sample fractionation (Hansson et al, 2017). We identified 33,605 peptides in CSF samples. Our study nearly doubled the number of reported peptide IDs in CSF of neurodegenerative disease patients without sample fractionation and, for the first time, we report the identification of tens of thousands of peptides in CSF of ALS patients. Thus, even after the removal of proteins, CSF samples remain complex, with peptide abundance dynamic range reaching eight orders of magnitude. The high number of peptide IDs was achieved through the use of state-of-the-art MS instrumentation, improvements in software packages, and systematic optimization of the sample preparation protocol and LC-MS parameters specifically for peptidomic analysis of CSF samples. Searching the peptidomics

data against peptide database did not significantly improved the number of identified peptides, demonstrating the comprehensiveness of our initial search against the protein database. Peptidomics promises to be a valuable tool for differential diagnosis, as demonstrated by hierarchical clustering analysis that clearly groups ALS patients and controls, and PCA that effectively distinguishes the two patient groups.

The main limitations of this study are the lack of direct experimental validation of the independent behavior of peptides from their parent proteins and unclarity whether these candidate biomarkers can reliably predict disease progression or be effectively used for staging and longitudinal monitoring.

In conclusion, we here identified several novel and promising peptide biomarker candidates in CSF for ALS using peptidomics. The peptide candidates are derived from proteins with different function and their determination with our PRM method might provide the opportunity for simultaneous investigation of key processes in ALS where no biomarkers are available so far. All peptides combined showed a better diagnostic performance than NFL and, in contrast to NFL, they seem to be ALS-specific and cover additional key processes in ALS. The NFL peptide PRM might be used as reference method for NFL immunoassays Nevertheless, future studies must clarify their clinical value for (differential) diagnosis, monitoring disease progression and treatment response, their changes in the preclinical phase of ALS and whether the peptide level changes are indicative of the parent protein concentration or reflect changes in protein processing.

## Methods

**Reagents and tools table**

| Reagent/resource | Reference or source | Identifier or catalog number |
| --- | --- | --- |
| **Chemicals, enzymes, and other reagents** | | |
| Non-human standard peptides | Peptide Specialty Laboratories | Custom-synthesized |
| Stable isotope-labeled standard peptides | Peptide Specialty Laboratories | Custom-synthesized |
| Trypsin/LysC | Promega | V507B |
| **Software** | | |
| PEAKS Studio v10.6 | https://www.bioinfor.com (Zhang et al, 2012) | |
| Perseus v1.6.15.0 | https://maxquant.net/perseus (Tyanova et al, 2016) | |
| Skyline v22.2 | https://skyline.ms/project/home/begin.view (MacLean et al, 2010) | |

| Reagent/resource | Reference or source | Identifier or catalog number |
|---|---|---|
| STRING v11.5 | https://version-11-5.string-db.org (Szklarczyk et al, 2023) | |
| R v4.3.1 | https://cran.r-project.org (RCoreTeam, 2024) | |
| R package: ggplot2 | Wickham, 2016 | |
| R package: ComplexHeatmap | Gu, 2016, 2022 | |
| R package: GGally | Schloerke et al, 2024 | |
| R package: DescTools | Signorell A, 2024 | |
| R package: factoextra | Kassambara and Mundt, 2020 | |
| R package: pROC | Robin et al, 2011 | |
| R package: nnet | Venables and Ripley, 2002 | |
| R package: ggcorrplot | Kassambara A, 2023 | |
| **Other** | | |
| Microcon - 30 kDa Centrifugal Filters (MWCO) | Merck, Germany | MRCF0R030 |
| AttractSPE Disks Bio SDB | Affinisep, France | #SPE-Disks-Bio-DVB-47.20 |
| Acclaim PepMap 100 trap column | Thermo | PN 164535 |
| Acclaim PepMap analytical column | Thermo | PN 164945 |
| FAIMS Orbitrap Exploris 480 | Thermo | |
| UltiMate 3000 RSLCnano | Thermo | |
| Ella Human NF-L Kit | ProteinSimple | |

## Methods and protocols

### Patients

Patients were recruited at the Department of Neurology, Ulm University Hospital. ALS was diagnosed according to the revised El Escorial criteria (Brooks et al, 2000), and all ALS patients were tested for known ALS gene mutations. The other neurodegenerative diseases included in the study were diagnosed according to established criteria (Emre et al, 2007; Hughes et al, 1992; McKeith et al, 2005; McKhann et al, 2011; Rascovsky et al, 2011). Controls in discovery, validation and neurodegeneration cohorts were age- and sex-matched individuals without neurodegenerative disease. All patients gave written informed consent to be included in this study, and the experiments conformed to the principles set out in the WMA Declaration of Helsinki and the Department of Health and Human Services Belmont Report. The Ethics Committee of Ulm University approved the study (approval no. 20/10). Demographic characteristics of the cohorts are listed in Table 1. CSF was collected by lumbar puncture during diagnostic workup, centrifuged and stored at −80 °C within 2 h in polypropylene tubes. The number of patients included in all cohorts was based on our previous experience with explorative biomarker studies.

### Sample preparation for peptidomic analysis

For the screening analysis, 200 µL CSF samples were spiked with 40 µL of a solution containing TEAB (triethylammonium bicarbonate), TCEP (tris(2-carboxyethyl) phosphine hydrochloride), CAA (2-chloroacetamide), and standard peptides (StdPep or SISPep), giving a final concentration of 100 mM TEAB, 1 mM TCEP and 1 mM CAA (Fig. 1A). Proteins and peptides were reduced and alkylated by incubating the sample for 10 min at 95 °C and 400 rpm. Afterwards, peptides were separated from proteins using 30 kDa molecular weight cut-off filter (MWCO) and by centrifugation at $11,000 \times g$. Isolated peptides were purified using STAGE Tips (Affinisep SPE-Disks-Bio-DVB-47.20) and eluted with 100 µL of 60% acetonitrile (ACN) and 0.1% formic acid (FA). The purified peptides were dried overnight in a SpeedVac and reconstituted in 13.5 µL 2% ACN, 0.5% trifluoroacetic acid (TFA). For the mass spectrometry (MS) analysis 10 µL of the reconstituted sample were injected into the chromatographic system (Fig. 1A).

The same sample preparation protocol was applied for the targeted, label-free parallel reaction monitoring (PRM) analysis of selected samples from the discovery cohort. However, for the PRM analysis of samples from the validation cohort, SISPep were added instead of StdPep. The methylated and isotope-labeled MYL1 standard peptide was added additionally to the SISPep to the samples from neurodegeneration cohort.

For high-pH fractionation, the same type of STAGE Tips were used as for the purification of peptides. The peptides were eluted in three fractions, each with 60 µL of 6%, 24% and 70% ACN solution in 20 mM ammonium formate and pH 10.

ALS and Con samples from the discovery cohort were measured alternately, while samples from the validation and neurodegeneration cohorts were randomized. The sample number in the source data for validation and neurodegeneration cohorts represent the order how they were measured. Despite diagnostic status for the systematic randomization, the analysts were blinded to the patient data until measurements were finished.

### Screening analysis

The screening analysis was performed with an UltiMate 3000 RSLCnano system and an Orbitrap Exploris 480 mass spectrometer equipped with a high-field asymmetric waveform ion mobility spectrometry (FAIMS) interface for online peptide fractionation according to their ion mobility properties. Peptides were loaded on a Thermo Acclaim PepMap 100 trap column (75 µm × 2 cm, 3 µm, 100 A, C18) by applying a flow rate of 5 µL/min, while for peptide separation a Thermo Acclaim PepMap analytical column (50 µm × 50 cm, 2 µm, 100 A, C18) was used at a flow rate of 150 nL/min. Peptides were eluted with a step-gradient from 1% to 53% B in 165 min and a total run time of 195 min. Mobile phase of the loading pump (trap column) was 0.05% TFA/2%MeOH, and of the nano pump (analytical column) 4% DMSO/0.1% formic acid (A) and 4%DMSO/76% acetonitrile/0.1% formic acid (B).

The eluted peptides were positively ionized using a stainless-steel emitter within a Nanospray Flex Ion Source by applying a spray voltage of 2000 V. FAIMS was set to operate at compensation voltages of −40, −50, and −65 V and the RF to 50%. The EASY-ICTM was used for internal mass calibration and the Run Start as calibration mode. A top-eight method was implemented for the data acquisition. Thereby, the MS spectra were acquired in a scan range of 400–1400 $m/z$ and a resolution of 120,000, 300% AGC and maximum injection time of 100 ms. For the acquisition of fragment spectra (MS/MS), peptide ions

were isolated with a quadrupole window of 1.6 $m/z$, 1 000% AGC and maximum injection time of 80 ms. Isolated ions were fragmented with a normalized HCD collision energy of 28% and the fragment spectra were recorded at a resolution of 30,000.

The PEAKS software was used for peptide identification and quantification. The parent and fragment mass error tolerances were set to 15.0 ppm and 0.02 Da. The settings for enzyme was "none", the digestion mode was "unspecific", and we searched for peptides with a length up to 65 amino acids. Carbamidomethylation was set as fixed modification and protein N-terminal acetylation and methionine oxidation as variable modifications. A maximum of five PTMs per peptide were allowed. The human reference proteome from UniProt (downloaded 12-Dez-2022) was used for peptide identification with a peptide and protein FDR of 1%. The FDR estimation was performed with the decoy fusion method. In the quantitative analysis of screening data, the mass error tolerance was set to 15.0 ppm and FDR to 1%.

For further quantitative analysis, we considered only peptides with a quality ≥ five and those identified in at least one sample per group. Peak area was used for quantification, and the data were normalized to five StdPep and exported from PEAKS. The statistical analysis of the LFQ data was conducted in Perseus. Here, after log2 transformation of the data, only peptides identified in at least 70% of control or ALS samples were taken into consideration for statistical analysis. Missing values were replaced by imputation from a normal distribution (width 0.3, down shift 1.8). A two-tailed Student's $t$ test was performed for group comparison analysis and correction for multiple testing was done with permutation-based FDR (0.05) and for the data visualization in the volcano plot the s0 was set to 0.1.

## Targeted PRM analysis

The PRM analysis was performed with the same LC-MS equipment, trap column, solvent composition and flow rate parameters as used in the screening analysis, but with a Thermo Acclaim PepMaP, 50 μm × 15 cm, 2 μm, 100 A, C18 analytical column and a gradient of 1 to 50% B in 40 min for peptide separation. The quadrupole isolation window was set to 1 $m/z$, RF to 50% and AGC to 3000%. All other peptide-specific parameters for the PRM method are listed in Table EV1. Skyline software was used for the analysis of PRM data. The data were normalized to respective SISPep reported as light to heavy ratio (L/H ratio) and to StdPep used as global standards (ratio to StdPep) in label-free PRM experiments.

## Development of a label-free PRM method

The label-free PRM method was developed using results from screening experiments and CSF pool samples instead of synthetic standard peptides. The list of peptide candidates for the development of the label-free PRM method included peptides found to be significantly regulated in screening experiments (Dataset EV2). To include as many peptides as possible five comparative analyses were performed with five different peptide selection criteria. This time for the group comparison statistical analysis were considered peptides quantified in at least: three samples per group with no data imputation, three control or ALS samples with no data imputation, 70% of the samples in each group with data imputation, 70% of

control or ALS samples and data imputation, 70% of all samples and data imputation. In addition, a second search was conducted using screening data, incorporating only carbamidomethylation as a fixed modification and methionine oxidation as a variable modification. This search also comprised three QC samples. Again, the same peptides selection criteria were applied for the group comparison statistical analysis of the data.

Peptides significantly regulated in ALS compared to controls from the first and second searches were then combined. Peptides that were shown to be unstable after sample freezing/thawing in screening experiments (with a variation over 100%) were excluded from the list. Many of these peptides were derived from CO3 protein. In addition, two albumin peptides from the list were excluded because they are blood-derived and four keratin peptides as potential contaminants. The final list consisted of 71 peptides.

All identified precursors of a peptide were included in the first PRM measurements through multiple injection of two CSF pool samples. A spectral library was built in Skyline with the PEAKS results file from the screening data. The peak picking in PRM data was based on transition co-elution, retention time comparison with screening data, and dotp value higher than 0.7 with at least 10 transitions. For the remaining peptides and after selection of best transition, FAIMS-compensation voltage and collision energy were optimized

In the next step, peptides underwent an analytical evaluation to assess peptide stability by applying up to five freeze–thaw cycles to the sample, dilution stability by up to 1 to 8 dilution of the sample and assay variation. Only peptides exceeding the control limit of ±30% variation were excluded from the final label-free PRM method, which consisted of 41 peptides and included five StdPep. The label-free PRM method was applied to selected samples from the discovery cohort, and the obtained results were validated by comparing with screening data in terms of up- and downregulation.

## Development of a PRM method

Samples from the validation cohort were measured by a PRM method by using SISPep. Peptides included in the PRM method were selected based on their performance in the label-free PRM. From the 11 peptides found significantly regulated in the label-free PRM data, we excluded several CO3 peptides because they were not stable in CSF. Neurofilaments (light, medium and heavy chains) are already well-established biomarkers in ALS, and we decided to focus on NFL peptides only as a proof of concept. Beyond the candidate peptides from the label-free PRM, we added a second MAP1B peptide to increase the coverage of this protein. For all peptides, the best transitions were selected, and both the FAIMS-compensation voltage and collision energy were optimized. After analytical evaluation of the PRM regarding stability upon freezing and thawing of the samples, dilution stability and assay variation, the APOC1_TP peptide was removed as it did not pass the dilution stability test. The final PRM method consisted of eight peptides and their respective SISPep (Table EV1).

## NFL determination by Ella

NFL in CSF was measured with an automated microfluidic immunoassay (Ella Human NF-L Kit from ProteinSimple, San

Jose, CA, USA) according to the manufacturer's instructions (Oeckl et al, 2023).

## Statistics

Statistical analysis was performed in R (v. 4.3.1). Data visualization was created using the ggplot2 package (v. 3.4.4), except for the Heatmap where the ComplexHeatmap package (v. 2.16.0) was used by applying the ward.D2 as clustering method. We calculated the %CV to evaluate the reproducibility of peptidomic and targeted MS measurements. Multi scatterplot plot for correlation analysis was created with the GGally package (v. 2.2.0) and the Spearman rank correlation coefficient. Data analysis for the volcano plot was performed in Perseus and results visualized with ggplot2. Sex difference in groups was tested with chi-squared test and for continues variables, group comparisons were performed with the Wilcoxon test (two groups) and Kruskal–Wallis test corrected with Dunn´s post hoc test for multiple comparison (three groups). Thereby, the packages stats (v. 4.3.1) and DescTools (v. 0.99.50) were used. PCA analysis was performed with stats (v. 4.3.1) and factoextra (v. 1.0.7) packages where the ellipse was generated with 95% confidence interval. Receiver operating characteristic (ROC) curves were generated with the package pROC (v. 1.18.5). For the ROC curve with all peptides a multinomial logistic regression was implemented using nnet package (v. 7.3-19) and the multinomial log-linear model. The ROC curve was then created with predicted probabilities from the model. The function cor from the stats package was used to calculate the Spearman's correlation coefficient between eight peptides and the correlation matrix was visualized with ggcorrplot package (v. 0.1.4.1). In the MCL clustering analysis performed on STRING (v. 11.5, June 2023) the inflation parameter was set to 3 and confidence score to medium. In addition, were selected full STRING as network type, evidence for meaning of network edges, all interactions sources, and none for max number of interactors to show.

---

**The paper explained**

**Problem**
Only few biomarker candidates are available so far for amyotrophic lateral sclerosis (ALS) and peptide levels in cerebrospinal fluid (CSF) are potential biomarker candidates but have not yet been studied at a large scale.

**Results**
Here, we used peptidomics in CSF of ALS patients to answer the question whether there are ALS-related changes of specific peptides. The discovery cohort identified 33,605 peptides. In a validation cohort, the levels of eight peptides derived from NFL, MAP1B, MYL1, APOC1, CADM3, SCG1, and PENK proteins changed in ALS compared to controls. Only the NFL peptide was significantly altered in other tested neurodegenerative diseases. Thus, we demonstrate the complexity and suitability of the CSF peptidome for biomarker research and show specific ALS-related changes representing key pathological processes in ALS and which can be used as novel biomarker candidates in ALS.

**Impact**
Our biomarker candidates describe key processes in ALS and other neurodegenerative diseases, and therefore, they might help in differential diagnosis, prognosis, monitoring of disease progression and treatment response.

---

## Graphics

Figure 1A and some of the synopsis graphics were created with BioRender.com.

## Data availability

The datasets produced in this study are available in the following databases: mass spectrometry data: ProteomeXchangeConsortium via PRIDE PXD062419. In the discovery cohort, samples from ALS patients are labeled with odd numbers in the MGF and RAW files, whereas in the result files (MZID and DiscoveryCohort_LFQ), they are numbered sequentially from 1 to 24.

The source data of this paper are collected in the following database record: biostudies:S-SCDT-10_1038-S44321-025-00272-w.

## Peer review information

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

## Acknowledgements

We are grateful to all patients for their participation in this study. We would like to thank Stephen Meier for his excellent technical assistance with Ella measurement and the biobank of the Department of Neurology in Ulm (Alice Beer, Sandra Hübsch and Dagmar Schattauer) for their help with providing the samples. No funding was received towards this work.

## Author contributions

**Besnik Muqaku**: Conceptualization; Data curation; Software; Formal analysis; Investigation; Visualization; Methodology; Writing—original draft; Writing—review and editing. **Johannes Dorst**: Investigation; Writing—review and editing. **Maximilian Wiesenfarth**: Investigation; Writing—review and editing. **Markus Otto**: Investigation; Writing—review and editing. **Albert C Ludolph**: Investigation; Writing—review and editing. **Patrick Oeckl**: Conceptualization; Supervision; Methodology; Project administration; Writing—review and editing.

Source data underlying figure panels in this paper may have individual authorship assigned. Where available, figure panel/source data authorship is listed in the following database record: biostudies:S-SCDT-10_1038-S44321-025-00272-w.

## Funding

## Disclosure and competing interests statement

PO received research support from the Cure Alzheimer Fund, ALS Association (24-SGP-691, 23-PPG-674-2), ALS Finding a Cure, the Charcot Foundation, the DZNE Innovation-to-Application program and consulting fees from LifeArc and Fundamental Pharma.

# Expanded View Figures

**Figure EV1.  Characterization of the MYL1 peptide.**

Targeted PRM chromatograms of a CSF pool sample spiked with the MYL1 stable isotope-labeled standard peptides (SISPep). Acetylated MYL1 SISPeps (blue, top and in the middle) eluted almost 1.5 min later than the endogenous MYL1 peptide (red). The trimethylated MYL1 SISPep (blue, bottom) eluted exactly at the same time as endogenous MYL1 peptide (red) and confirmed the identity of MYL1 peptide.

▶

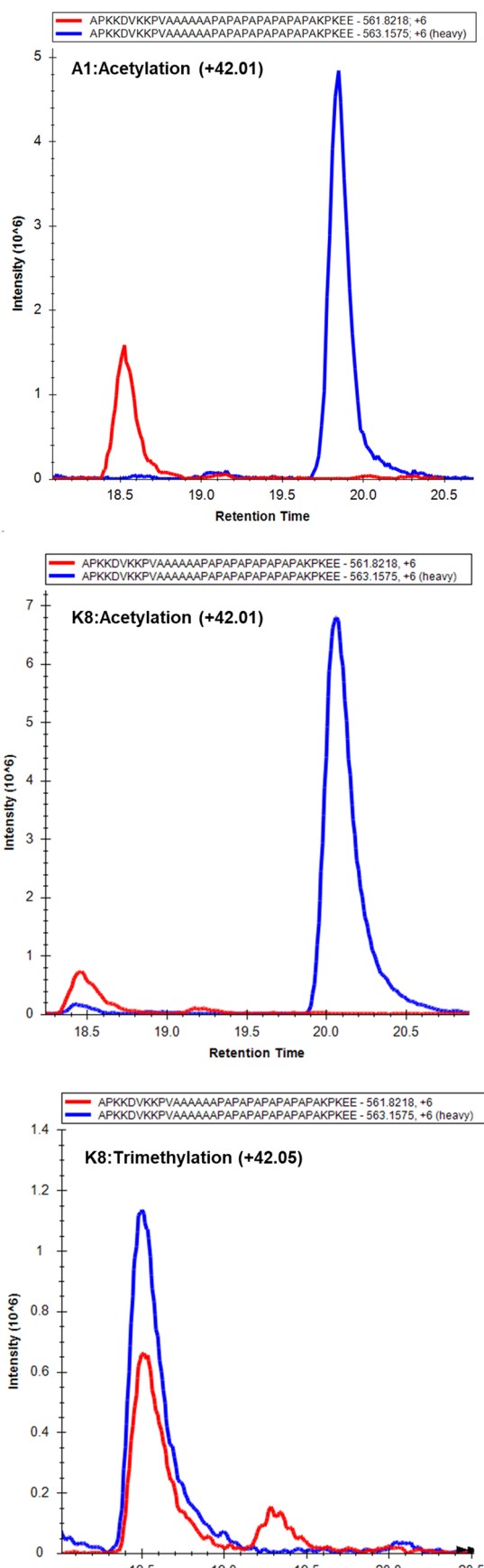

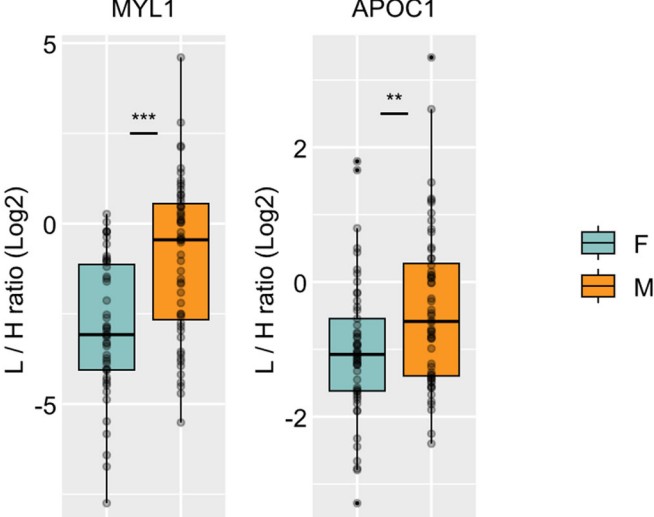

**Figure EV2. Levels of MYL1 and APOC1 in females and males.**

Boxplots show the PRM data for MYL1 and APOC1 by comparing females ($n = 53$) and males ($n = 56$) over all patients from validation cohort. For both peptides the data were normalized to their respective stable isotope-labeled standard peptide (SISPep) indicated as light to heavy ration (L/H ratio). Wilcoxon test was applied for group comparison analysis. Data are visualized by boxplots (minima, maxima, center, bounds of box and whiskers, and percentile) and individual values. Peptide (log2 fold change; *P* value): MYL1(1.97; $1.3*10^{-5}$) and APOC1(0.60; 0.0097). Fold change (average M (log2)-average F (log2)). Log2 fold change >0 indicates an increase in males. **P value < 0.01; ***P value < 0.001.

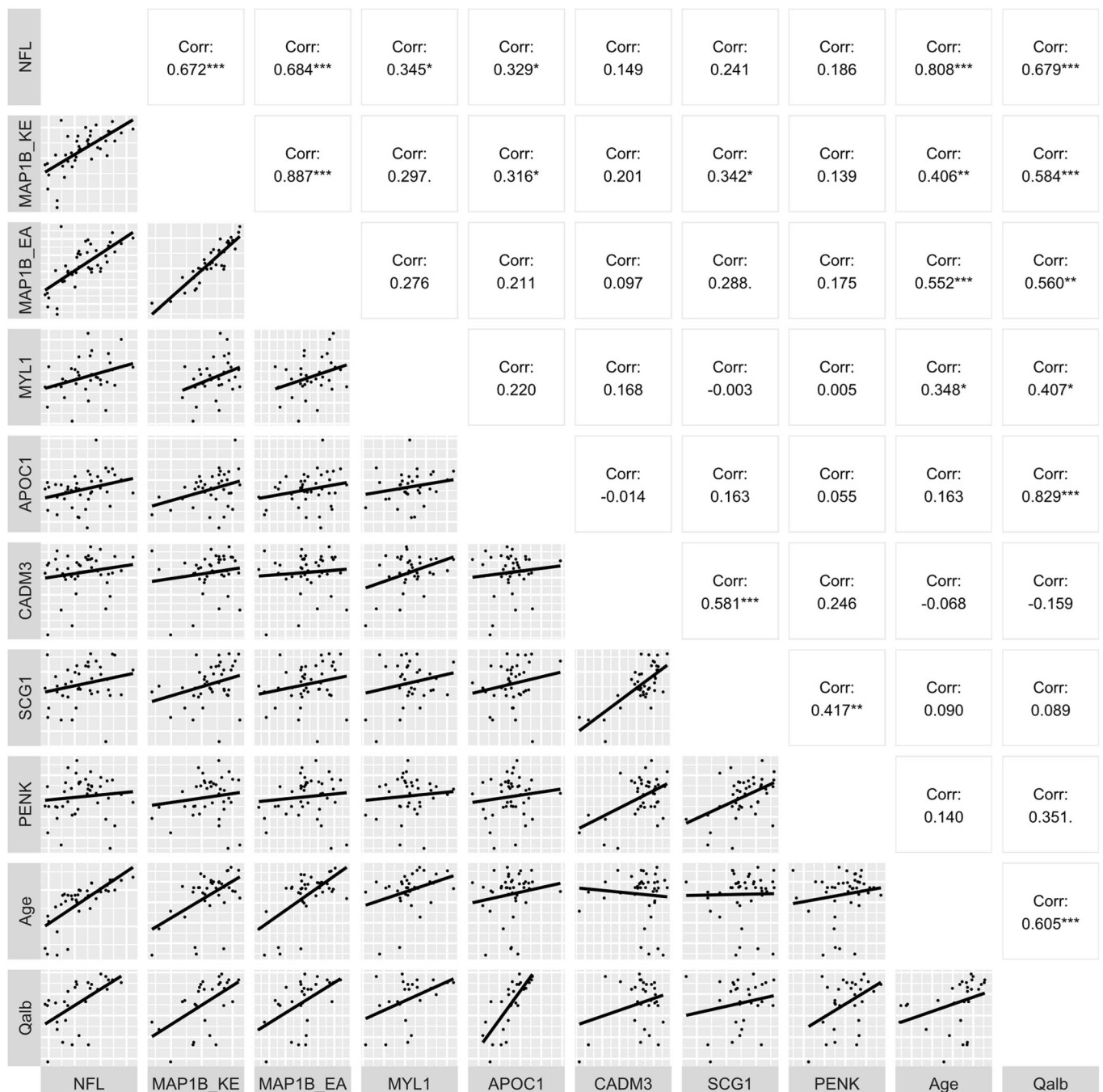

**Figure EV3. Correlation of peptides with clinical data in controls.**

Multi scatterplot show the correlation and Spearman's correlation coefficient for eight peptides and clinical data only in controls from the validation cohort. All data were log2 transformed and each dot represents the data of a patient. The two different MAP1B peptides are indicated by the first two amino acids of the peptide sequence. Qalb - albumin quotient. *P value < 0.05; **P value < 0.01; ***P value < 0.001.

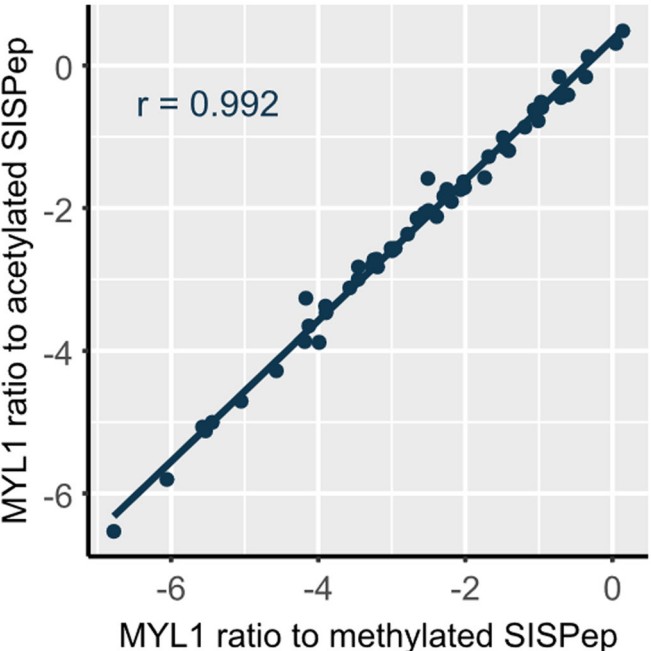

**Figure EV4.  Correlation of MYL1 normalized to acetylated and methylated standard.**

Scatterplot show the correlation and Spearman's correlation coefficient for the ratio of MYL1 peptide normalized to acetylated and methylated stable isotope-labeled standard peptide (SISPep) in samples from neurodegeneration cohort. Log2 data.

