## [Peer Review File · EMBO Molecular Medicine]

Peptidomic analysis of CSF reveals new biomarker candidates for amyotrophic lateral sclerosis

Besnik Muqaku, Johannes Dorst, Maximilian Wiesenfarth, Markus Otto, Albert Ludolph, and Patrick Oeckl

Corresponding author: Patrick Oeckl (patrick.oeckl@uni-ulm.de)

Review Timeline:

Submission Date:	4th Feb 25
Editorial Decision:	27th Feb 25
Revision Received:	19th May 25
Editorial Decision:	11th Jun 25
Revision Received:	26th Jun 25
Accepted:	3rd Jul 25

Editor: Zeljko Durdevic

Transaction Report:

27th Feb 2025

Dear Dr. Oeckl,

Thank you for the submission of your manuscript to EMBO Molecular Medicine. We have now received feedback from the three reviewers who agreed to evaluate your manuscript. All three referees recognize interest of the study but also raise important and partially overlapping concerns that should be addressed in a major revision. Focus of the revision should be in comparing the peptide panel to other neurodegenerative diseases and to ASO-treated SOD1-ALS patients using published datasets as suggested by the referees. If you would like to discuss further the points raised by the referees, I am available to do so via email or video. Let me know if you are interested in this option.

We would welcome the submission of a revised version within three months for further consideration. Please let us know if you require longer to complete the revision.

I look forward to receiving your revised manuscript.

Yours sincerely,

Zeljko Durdevic

We require:

- 1) A .docx formatted version of the manuscript text (including legends for main figures, EV figures and tables). Please make sure that the changes are highlighted to be clearly visible.
- 2) Individual production quality figure files as .eps, .tif, .jpg (one file per figure). For guidance, download the 'Figure Guide PDF': (<https://www.embopress.org/page/journal/17574684/authorguide#figureformat>).
- 3) A .docx formatted letter INCLUDING the reviewers' reports and your detailed point-by-point responses to their comments. As part of the EMBO Press transparent editorial process, the point-by-point response is part of the Review Process File (RPF), which will be published alongside your paper.
- 4) A complete author checklist, which you can download from our author guidelines (<https://www.embopress.org/page/journal/17574684/authorguide#submissionofrevisions>). Please insert information in the checklist that is also reflected in the manuscript. The completed author checklist will also be part of the RPF.
- 5) Please note that all corresponding authors are required to supply an ORCID ID for their name upon submission of a revised manuscript.

6) It is mandatory to include a 'Data Availability' section after the Materials and Methods. Before submitting your revision, primary datasets produced in this study need to be deposited in an appropriate public database, and the accession numbers and database listed under 'Data Availability'. Please remember to provide a reviewer password if the datasets are not yet public (see <https://www.embopress.org/page/journal/17574684/authorguide#dataavailability>).

12) Author contributions: You will be asked to provide CRediT (Contributor Role Taxonomy) terms in the submission system. These replace a narrative author contribution section in the manuscript.

13) A Conflict of Interest statement should be provided in the main text.

14) Every published paper now includes a 'Synopsis' to further enhance discoverability. Synopses are displayed on the journal webpage and are freely accessible to all readers. They include a short stand first (maximum of 300 characters, including space) as well as 2-5 one-sentences bullet points that summarizes the paper. Please write the bullet points to summarize the key NEW

findings. They should be designed to be complementary to the abstract - i.e. not repeat the same text. We encourage inclusion of key acronyms and quantitative information (maximum of 30 words / bullet point). Please use the passive voice. Please attach these in a separate file or send them by email, we will incorporate them accordingly.

15) Include a Reagents and Tools Table as part of the Methods section, which can be downloaded from our author guidelines (<https://www.embopress.org/page/journal/17574684/authorguide#structuredmethods>)

**** Reviewer's comments ****

Referee #1 (Remarks for Author):

The manuscript is generally very informative and interesting for the scientific community, especially with regard to biomarker identification for ALS. ALS is one of the rare diseases in which the motor neurons are affected, which leads to muscle weakness as the disease progresses and to death in most patients after a few years. Diagnosis is currently limited to clinical diagnosis and new (molecular) diagnostic biomarkers are therefore urgently needed.

The authors describe a study exploring the peptidome of ALS patients in CSF. They investigated CSF samples of ALS and non-degenerative control patients. The study design is very advanced (in the field of proteomics and peptidomics), as the total cohort was divided into a discovery (n=48) and a validation cohort (n=109) to generate results that are as valid as possible. In addition, attention was at least paid to matching the disease and control cohorts in terms of gender and age.

The samples were analysed using standardised mass spectrometric methods for peptidome analysis. In total 33605 peptides were identified, from which it was possible to establish a panel of 8 peptides of 7 proteins (NFL, MAP1B, MYL1, APOC1, CADM3, SCG1, PENK) discriminating ALS patients from the respective controls.

Such a study has so far not been published (only a preprint exists, <https://doi.org/10.1101/2023.12.14.23299946>). Therefore, the results will be of interest for a wide research community, since even for other neurodegenerative diseases, such studies are currently very rare.

Although the research work and the results are new and highly relevant and the study design as such is convincing, I still have the following critics that should be addressed and answered before the article may be published in EMBO Molecular Medicine:

Major critics:

1. The provided supplementary data is far from being sufficient. Neither the raw files from the MS measurements, nor the complete data regarding the peptide identification and quantification in all the measurements is provided. This data would however be essential for the scientific community. Currently, the quality of the obtained data cannot be investigated at all.
2. Performance of the resulting peptide panel was not compared to other neurodegenerative diseases. Thus for me it is questionable how applicable it will be in clinical routines. It is yet to be determined if the assay is specific for ALS or neurodegeneration in general (for example: peptides of proenkephalin-A and secretogranin-1 were found to be significantly decreased in AD patients compared to CTRLs; <https://doi.org/10.1021/pr501076j>). It might have been useful to include other cohorts of neurodegenerative diseases. This will be difficult in retrospect, but at least the results should be compared with those already known for other neurodegenerative diseases. There may even be data sets in public repositories that could be used directly for this purpose.
3. Was blood-contamination investigated in CSF samples? In our experience, the effect of blood contamination in the CSF samples taken is very high and blood contamination should be ruled out. In our experience, the effect of blood contamination in the CSF samples taken is very high and blood contamination should be ruled out at all costs. In retrospect, it could be helpful to evaluate published data in this regard and compare it with the present data.
4. In the results section, it does not become clear what the newly established method offers in terms of novel diagnostic options, since it is said that results correlate very well with an established ELISA assay for NFL. It is not stated how the presented method performs compared to ELISA. The value of this newly established method should be clearly stated.
5. The data analysis is not optimal:
 - a) No specific peptide database was used, which is recommended for peptidomic analyses. (<https://academic.oup.com/database/article/doi/10.1093/database/baae113/7887558>; <https://doi.org/10.1038/s43586-023-00205-2>). The used Uniprot FASTA file does not include the required information on potential cleavage sites. Therefore, the obtained raw files should be reanalyzed using a specific peptidome database and the digestion mode in PEAKS should be set to "no digestion". Received data should then be compared to the original peptide list.
 - b) It does not become completely clear, how the quality of the data was assessed, since there are no comments on the use of decoy peptides during data evaluation.
 - c) Protein interference is not necessary for the peptidomics approach, thus the filtering based on 1% FDR on the protein level may have reduced the number of potentially suitable peptides before data evaluation. The obtained data should therefore be

reanalyzed and results should be compared to the original results, to check for additional candidates.

Minor critics:

6. Page 8: please rephrase "Additionally, from the list were excluded two albumin peptides because they are blood-derived" to "Additionally, two albumin peptides from the list were excluded because they are blood-derived".
7. Page 8 middle, section "Development of a label-free PRM method): selection criteria for used peptides are not clear, are these different criteria or is there some kind of grading involved?
8. Page 9: please rephrase "For all peptides, best transitions were selected as well as FAIMS - compensation voltage and collision energy were optimized" for better readability.
9. Page 10: Please reference papers regarding R and software packages.
10. Page 11: Please clarify the sources/constitution for the QC samples.
11. Page 12: Figure 2 F does not appear to be created in Perseus as mentioned in the "Material and Methods" part.
12. Page 12: Why was the label-free PRM method applied to 10 and 12 samples and which criteria were applied to select the used samples?
13. Page 13: Please reference the publication on STRING DB and indicate the settings used to generate respective networks in the methods section.
14. Discussion section is way to long and should be shortened. In addition, the conclusion in the end seems a bit too speculative.
15. Page 33 Figure 4D: A ROC curve should also be displayed for the ELISA, as it cannot be evaluated if the PRM panel provides a benefit compared to established analytical methods.
16. Page 33 Figure 4 E: The correlation with ELISA results should also be checked with determined concentrations from the PRM assay, as these should be similar when the same samples were used.
17. Please add a paragraph describing the ELISA measurements in the material and methods section.

Referee #2 (Comments on Novelty/Model System for Author):

While the study offers valuable insights, it lacks major conceptual breakthroughs as many markers identified in the panel, particularly NFL, are heavily used in clinical assessment already. Further validation in other biofluids, such as serum, might strengthen the general assessment of ALS in the real world as it is unlikely people will screen CSF regularly for ALS onset and progression unless they are undergoing some ASO/gene therapy treatments.

Referee #2 (Remarks for Author):

The manuscript titled "Peptidomic analysis of CSF reveals new biomarker candidates for amyotrophic lateral sclerosis" by Muqaku and colleagues explores potential peptide biomarkers in cerebrospinal fluid (CSF) for amyotrophic lateral sclerosis (ALS). As one of the biggest challenges for ALS diagnosis is the lack of reliable biomarkers and how to assess the severity of disease progress, the identification of bona fide disease onset and progression prediction biomarkers is an essential and critical research direction in ALS. The authors used peptidomic analysis to identify dysregulated peptides in ALS patients, optimizing normalization conditions for mass spectrometry screening. They identify candidates in a discovery cohort (n = 48) and validate them via label-free PRM analysis in an independent cohort (n = 109). A logistic regression model combining upregulated (NFL, MAP1B, MYL1, APOC1) and downregulated (CADM3, SCG1, PENK) peptides achieves an AUC of 98%, highlighting strong diagnostic potential. While the study offers valuable insights, it lacks major conceptual breakthroughs as many markers identified in the panel, particularly NFL, are heavily used in clinical assessment already. Further validation in other biofluids, such as serum, might strengthen the general assessment of ALS in the real world as it is unlikely people will screen CSF regularly for ALS onset and progression unless they are undergoing some ASO/gene therapy treatments.

Major points:

1. The current layout of the manuscript appears to primarily emphasize the procedure of peptidomics analysis in the CSF of ALS patients. If the focus is on methodology, what distinguishes this approach from previous studies? Alternatively, if the goal is to identify novel biomarkers, the authors should provide stronger evidence or clarify the novelty of their findings. While the study selects several peptide biomarker candidates, many have already been proposed as ALS biomarkers, serving as a proof of principle. This reinforces the value of quantitative peptidomic analysis for studying disease biomarkers but does not fully support the manuscript's claim of discovering truly "new" biomarkers. Strengthening the evidence or highlighting the novelty of the identified biomarkers would enhance the study's impact.
2. Some rationales and descriptions in this study are unclear and could benefit from additional information to clarify their biological significance or underlying mechanisms. Here are a few examples:
 - a) Although the coefficient of variation (CV) is a common metric for assessing reproducibility in proteomics, providing more details in the Results or Methods section would help readers better understand its implications.

In Figures 1 and 2, while StdPep improves CV for standard peptides compared to other groups, the overall CV for the StdPrep group appears slightly higher for all quantified peptides. This raises concerns about variability and reliability. Could the authors clarify this discrepancy?

b) The alteration of tryptic-like peptides in ALS may result from endogenous protease activity or proteolytic processing linked to disease progression. However, additional experiments, such as longitudinal studies, may be needed to evaluate peptide levels over time and clarify their role in disease progression.

3. I found it challenging to assess the biomarker authenticity at "one single time" point. While it might be challenging, assessing these markers from ASO-treated SOD1-ALS patients from different time points and comparing the biomarker changes and their correlations to the motor enhancement scores (ALSF_{RS}) might provide the best evidence for verifying the biomarker panels in this study. Similar studies were performed before such as PMID: 37064776.

Minor points:

1. Would it be possible for the authors to perform GO enrichment analysis to determine whether the identified dysregulated peptides in the ALS group are associated with specific biological processes?

2. In Figure 2G, the authors perform peptidomic screening analysis on the discovery cohort and observe a separation of ALS groups in the hierarchical clustering analysis. What is the primary factor driving this separation (e.g., disease subtype, severity, gender, age, etc.)? Additionally, it would be valuable to discuss potential differences in peptidomics across various ALS subtypes.

3. In Table 1, there are several suggestions as below:

a) I assume that genetic ALS (gALS) refers to familial ALS (fALS). In most literature, individuals with familial ALS who carry mutations in known ALS-associated genes are typically referred to as having fALS. Is there a specific reason for using "gALS" in this manuscript?

b) Although the authors separated the groups by gender, I was wondering about the female-to-male ratio within each genetic type under "gALS."

c) Providing abbreviations in the table helps readers understand the content. However, the order appears to follow an alphabetical arrangement rather than a logical flow. It may be more intuitive to present the abbreviations in the order they appear from row to column.

d) There should be a superscript "c" instead of "d."

2. In Table 2, what is the condition of evaluating the data using a shorter LC gradient? How does it differ from the regular conditions, and how should the results be interpreted?

3. There are some typos or inconsistent characters in the text and figures. Please review and correct them.

Referee #3 (Remarks for Author):

In this study, Muqaku et al. utilized mass spectrometry-based peptidomic analysis to examine cerebrospinal fluid (CSF) samples from ALS patients and non-neurodegenerative controls. From a discovery cohort, the authors identified eight candidate peptides. These peptides were further validated in a validation cohort, where the combination of these peptides achieved an area under the curve of 98% in receiver operating characteristic analysis, suggesting potential diagnostic utility. The study employs a systematic screening process and a rigorous quality control strategy, enhancing the reliability of its findings. Additionally, the detailed analysis of the data provides opportunities to explore key pathological processes in ALS and other neurodegenerative diseases. However, I do have some comments which the authors need to address for further strengthen this work

1. In this analysis, age- and sex-matched individuals without neurodegenerative diseases were used as controls. However, in clinical practice, ALS is often misdiagnosed or confused with other motor neuron diseases. Including patients with such conditions in the validation cohort could enhance the clinical translational potential of the study.

2. The study identified eight peptides as potential biomarkers for ALS, derived from seven different proteins. I am particularly interested in whether these source proteins are present in the CSF. If they are not found in the CSF, it would further underscore the significance of this peptidomic analysis. Conversely, if these proteins are present in ALS CSF, do their levels follow a similar trend as the identified peptides? Additionally, the study mentions that the peptide derived from MYL1 exists in a trimethylated form. I am curious whether this modification occurs on the intact MYL1 protein or if it arises post-cleavage at the peptide level.

3. The manuscript states that "PENK was especially down-regulated in C9orf72 mutation carriers even compared to sALS", suggesting potential differences in the CSF peptidome between sALS and gALS patients. Conducting a comparative analysis of data from sALS and gALS patients at the early experimental stage, particularly during the screening peptidomic analysis, could help identify additional differentially expressed peptides between the two groups. Such findings may offer valuable insights into the etiological differences between sALS and gALS.

4. The manuscript reports significant correlations among the levels of up-regulated peptides. I am curious whether this implies a functional or regulatory relationship among these peptides. Although the analysis indicates no significant correlation between individual peptide levels and FRS_r, could an integrated analysis of multiple peptides reveal a potential association with FRS_r?

5. I also have a few suggestions regarding the presentation of data and figures:

- (1) For correlation plots, such as Figures 1H and 4F, I recommend displaying the correlation coefficients directly on the scatter plots to enhance clarity. Additionally, in Figure 4F, I suggest grouping up-regulated and down-regulated peptides, as this would enhance readability and facilitate interpretation.
- (2) In the figure legends, the placement of panel labels is inconsistent, appearing before the legend in some cases and after it in others. I recommend standardizing the format for clarity and consistency.
- (3) Some abbreviations, such as FRS_r, are used in the manuscript without providing their full definitions. I recommend defining all abbreviations upon their first occurrence for clarity.
- (4) Certain analyses are discussed in the manuscript without the corresponding data being presented. I suggest either including the relevant data or revising the text to remove these references (e.g., Result Section 4, Paragraph 3).
- (5) I recommend reorganizing the Methods section to align with the order of the main text. This would improve readability and help readers better understand the role of each technique in the study.

Referee #1

Major critics:

1. The provided supplementary data is far from being sufficient. Neither the raw files from the MS measurements, nor the complete data regarding the peptide identification and quantification in all the measurements is provided. This data would however be essential for the scientific community. Currently, the quality of the obtained data cannot be investigated at all.

Response

Providing of data was not required at the first submission. Now we made mass spectrometry screening data generated with CSF samples accessible for the referees. Screening data have been deposited to the ProteomeXchangeConsortium with the identifier number PXD062419. The reviewer account details are Username: reviewer_pxd062419@ebi.ac.uk and Password: E98wnWOYHS4. In the discovery cohort, samples from ALS patients are labeled with odd numbers in the MGF and RAW files, whereas in the result files (MZID and DiscoveryCohort_LFQ), they are numbered sequentially from 1 to 24.

2. Performance of the resulting peptide panel was not compared to other neurodegenerative diseases. Thus for me it is questionable how applicable it will be in clinical routines. It is yet to be determined if the assay is specific for ALS or neurodegeneration in general (for example: peptides of proenkephalin-A and secretogranin-1 were found to be significantly decreased in AD patients compared to CTRLs; <https://doi.org/10.1021/pr501076j>). It might have been useful to include other cohorts of neurodegenerative diseases. This will be difficult in retrospect, but at least the results should be compared with those already known for other neurodegenerative diseases. There may even be data sets in public repositories that could be used directly for this purpose.

Response

This is a great idea which could contribute to further improve our manuscript. Therefore, we included to this manuscript targeted MS data generated from a neurodegeneration cohort which consisted of patients with Alzheimer's disease (AD), behavioral variant of frontotemporal dementia (bvFTD) and Parkinson's disease (PD) as well as control (Con) patients without neurodegenerative diseases (page 10, line 234). The results have been shown in a new figure (Figure 6).

Our peptide candidates derived from proenkephalin-A and secretogranin-1 were not reported to be regulated in the mentioned paper (<https://doi.org/10.1021/pr501076j>). MAP1B shows how important it is to look at peptide level since from ten quantified MAP1B peptides only two of them were significantly regulated and if we would do the analysis as in standard proteomics, where protein abundance is calculated as average of the abundance of unique peptides, then MAP1B as protein would not be increased in ALS (page 12, line 271).

3. Was blood-contamination investigated in CSF samples? In our experience, the effect of blood contamination in the CSF samples taken is very high and blood contamination should be ruled out. In our experience, the effect of blood contamination in the CSF samples taken is very high and blood contamination should be ruled out at all costs. In retrospect, it could be helpful to evaluate published data in this regard and compare it with the present data.

Response

All CSF samples were collected on a routine bases by the biobank of the Ulm University Hospital and underwent a standard quality control procedure of CSF labor diagnostic which includes also test for blood-contamination.

4. In the results section, it does not become clear what the newly established method offers in terms of novel diagnostic options, since it is said that results correlate very well with an established ELISA assay for NFL. It is not stated how the presented method performs compared to ELISA. The value of this newly established method should be clearly stated.

Response

The ROC curve generated with Ella data for NFL was added to the Figure 4D now enabling direct comparison with the NFL peptide. We included NFL in our analysis to enable the comparison of our findings with this established marker in ALS. However, the novelty of our findings is not NFL but the other peptides in our panel which have not been described before. The targeted MS method for the NFL peptide might however be used as a reference method for standardization of NFL measurements. A statement has been added in the conclusion part of the Discussion (page 18, line 432).

5. The data analysis is not optimal:
 - a) No specific peptide database was used, which is recommended for peptidomic analyses. (<https://academic.oup.com/database/article/doi/10.1093/database/baae113/7887558>; <https://doi.org/10.1038/s43586-023-00205-2>). The used Uniprot FASTA file does not include the required information on potential cleavage sites. Therefore, the obtained raw files should be reanalyzed using a specific peptidome database and the digestion mode in PEAKS should be set to "no digestion". Received data should then be compared to the original peptide list.
 - b) It does not become completely clear, how the quality of the data was assessed, since there are no comments on the use of decoy peptides during data evaluation.
 - c) Protein interference is not necessary for the peptidomics approach, thus the filtering based on 1% FDR on the protein level may have reduced the number of potentially suitable peptides before data evaluation. The obtained data should therefore be reanalyzed and results should be compared to the original results, to check for additional candidates.

Response

- a. Thank you for this important remark. We followed your suggestion and searched our data against a peptide database. Thereby, peptidatlas FASTA file from peptipedia 2.0 was used to create a peptide library in PEAKS. In 48 samples from the discovery cohort, we identified 2891 peptides (compared with 33605 in our original approach) and from them only three were significantly increased in ALS compared to controls. However, all three peptides were already contained in the list of identified and quantified peptides generated by our original search using the Uniprot FASTA. 217 peptides were identified only with the peptide database thus making only 0.6% of total peptides identified in the search with the Uniprot FASTA. These results show the comprehensiveness of our initial search against protein database and therefore, we decided to keep the original data.
- b. We used the decoy fusion method to estimate the false discovery rate (FDR) of peptide identifications. *Decoy fusion is an enhanced target-decoy method for result validation with FDR. Decoy fusion appends a decoy sequence to each protein as the "negative control" for the search.* In the method section of the manuscript the following is added: *The FDR estimation was performed with the decoy fusion method.*
- c. In PEAKS Xpro the FDR criteria can be changed after the search is finished and by setting FDR for protein groups at 100% the number of identified peptides did not change. We are aware that we may not have yet explored all the information contained in the raw data. However, once this manuscript is accepted, we will share the raw data with the scientific community (see point 1), allowing others to further investigate and uncover additional relevant insights.

Minor critics:

6. *Page 8: please rephrase "Additionally, from the list were excluded two albumin peptides because they are blood-derived" to "Additionally, two albumin peptides from the list were excluded because they are blood-derived".*

Response

The change has been implemented (page 24, line 559).

7. *Page 8 middle, section "Development of a label-free PRM method): selection criteria for used peptides are not clear, are these different criteria or is there some kind of grading involved?*

Response

These are five different peptide selection criteria and five different comparisons, including the main one showed in the volcano plot.

To make it clear the following is written in the manuscript: *... five comparative analyses were performed with five different peptide selection criteria* (page 23, line 546).

8. Page 9: please rephrase "For all peptides, best transitions were selected as well as FAIMS - compensation voltage and collision energy were optimized" for better readability.

Response

Thank you for this advice. The sentence is rephrased to: For all peptides, the best transitions were selected, and both the FAIMS - compensation voltage and collision energy were optimized (page 25, line 584).

9. Page 10: Please reference papers regarding R and software packages

Response

Added.

10. Page 11: Please clarify the sources/constitution for the QC samples.

Response

The QC sample is a CSF pool sample. This is described on page 6, line 131.

11. Page 12: Figure 2 F does not appear to be created in Perseus as mentioned in the "Material and Methods" part.

Response

Thank you for the advice that this is not clear. The data analysis was made in Perseus but the results visualized with R using ggplot2 package. To clarify it in the statistics part is added: ...and results visualized with ggplot2 (page 26, line 602).

12. Page 12: Why was the label-free PRM method applied to 10 and 12 samples and which criteria were applied to select the used samples?

Response

In this case, samples from the discovery cohort were selected based on availability because the screening data were used to validate the results of the label-free PRM analysis in terms of up-and down-regulation.

13. Page 13: Please reference the publication on STRING DB and indicate the settings used to generate respective networks in the methods section.

Response

The settings used in STRING clustering analysis are indicated in the statistics part of the Methods section and were removed from the figure legend (page 26, line 616). A STRING reference is added.

14. Discussion section is way to long and should be shortened. In addition, the conclusion in the end seems a bit too speculative.

Response

We agree with the reviewer that the Discussion section is quite long. This is because we think it is of great relevance for our findings to provide a brief discussion for each of the seven parent proteins of our peptide candidates for the readers of the manuscript. The Discussion is divided in several paragraphs making it easy to skip the discussion of individual peptides by the reader if desired. Since the length of the manuscript is within the allowed word count, we would like to keep the discussion of all peptides. Regarding the conclusions, we adjusted the conclusion part of the Discussion to make it less speculative (page 18, line 429).

15. Page 33 Figure 4D: A ROC curve should also be displayed for the ELISA, as it cannot be evaluated if the PRM panel provides a benefit compared to established analytical methods.

Response

The ROC curve generated with Ella data for NFL is added to the Figure 4 D.

16. Page 33 Figure 4 E: The correlation with ELISA results should also be checked with determined concentrations from the PRM assay, as these should be similar when the same samples were used.

Response

We performed only relative quantification with the PRM assay (light-to-heavy ratio) and there are no absolute concentration values available.

17. Please add a paragraph describing the ELISA measurements in the material and methods section.

Response

An Ella measurements paragraph has been added in the Methods section on page 25.

Referee #2

Major critics:

1. The current layout of the manuscript appears to primarily emphasize the procedure of peptidomics analysis in the CSF of ALS patients. If the focus is on methodology, what distinguishes this approach from previous studies? Alternatively, if the goal is to identify novel biomarkers, the authors should provide stronger evidence or clarify the novelty of their findings. While the study selects several peptide biomarker candidates, many have already been proposed as ALS biomarkers, serving as a proof of principle. This reinforces the value of quantitative peptidomic analysis for studying disease biomarkers but does not fully support the manuscript's claim of discovering truly "new" biomarkers.

Strengthening the evidence or highlighting the novelty of the identified biomarkers would enhance the study's impact.

Response

We thank reviewer #2 for the advice that the main goal of the study is not totally clear. Our main goal was to identify novel biomarker candidates for ALS. However, we wanted to use an optimized peptidomics workflow compared to previous studies enabling deeper peptidomic screening. This is why also some methodological optimization is described in the manuscript. To make our main goal clearer, we rephrased some parts in the Abstract, Discussion and Conclusion sections (pages 2,11,12, 18; lines 27,268, 427).

We identified eight novel peptide biomarker candidates in our study. We agree with the reviewer, that some (not all) of the respective parent proteins have already been described as ALS biomarker candidates, however, the peptides we identified here have not been described as ALS biomarkers before and are novel. The fact, that some parent proteins are already established markers such as NFL enables us to show the reliability of our study. On the other hand, our study clearly indicates that peptides and their parent proteins do not necessarily show similar changes indicated for example by MAP1B where some peptides were changed in ALS but others are not. So, our study provides not only novel peptide biomarker candidates but also information on which peptide sequences from a given protein are changed and which are not. In addition to novel peptides from previously described protein biomarkers, our final panel of peptide biomarkers also includes peptides which have not been described as peptide or protein biomarker in ALS before such as MYL1, MAP1B or PENK. In addition, as we recently reviewed (Muqaku et al., IJMS 2022) and also stated in the introduction, CSF peptidomic studies are rare in neurodegenerative diseases except Alzheimer's. We also added data of CSF levels of our peptide panel in other neurodegenerative diseases. Thus, there is a high degree of novelty in our study. We added several of the above statements in the Discussion section and also moved some paragraphs regarding importance to further clarify the novelty of our study (pages 12, 18; lines 268, 418, 428).

2. Some rationales and descriptions in this study are unclear and could benefit from additional information to clarify their biological significance or underlying mechanisms. Here are a few examples:

a) Although the coefficient of variation (CV) is a common metric for assessing reproducibility in proteomics, providing more details in the Results or Methods section would help readers better understand its implications.

In Figures 1 and 2, while StdPep improves CV for standard peptides compared to other groups, the overall CV for the StdPrep group appears slightly higher for all quantified peptides. This raises concerns about variability and reliability. Could the authors clarify this discrepancy?

b) The alteration of tryptic-like peptides in ALS may result from endogenous protease activity or proteolytic processing linked to disease progression. However, additional experiments, such as longitudinal studies, may be needed to evaluate peptide levels over time and clarify their role in disease progression.

Response

- a. We thank the reviewer for the suggestion and added some more information about the CV (page 5, line 121). Regarding the differences of the overall CV with the different normalization methods, we would first like to emphasize that the similarly good overall CV of the approach without normalization compared with the two normalization procedures underpins the high analytical quality and reproducibility of our measurements. The addition of internal standards in form of peptides such as StdPep is an established procedure in quantitative mass spectrometry. At first glance, the slightly higher CV for the StdPep argues against its use as a normalization strategy here. However, there are some specific reasons why the TIC approach and no normalization seem to be better in the repeated analysis of CSF QC samples: (1) The CVs were calculated from technical replicates of a CSF QC sample but are actually derived from the same sample and thus have exactly the same composition. Therefore, with a highly standardized and quality-controlled analysis as in our study, the CV is even low without any normalization. The technical replicates of the QC samples do, however, not reflect the effect of inter-individual variations in the CSF composition, matrix effects or other confounding factors from samples from different patients and it can be expected that in the QC samples the real CV in patient samples is underestimated. (2) A similar underestimation can be expected for the TIC approach because in the technical replicates an identical amount of peptides is present. In patient samples, the total peptide amount can vary considerably. This is also due to the physiology of CSF which is an ultrafiltrate of the blood and 80% of proteins in CSF are blood derived (and it can be expected that it is the same for their peptides) and concentrations depend on different factors e.g. CSF flow or blood-CSF-barrier integrity. Thus, they will behave different than brain-derived peptides. Thus, inter-individual variations in the CSF peptide content and composition (unrelated to the investigated disease) will significantly affect the TIC approach. The StdPep approach is independent from all these effects and the CV observed with the CSF QC samples can be expected to reflect also the situation in patient samples. In conclusion, we think the lower CV in QC samples without normalization and with the TIC approach compared with StdPep is due to the specific conditions accompanied with technical replicates but underestimates the CV that would be observed in samples from different patients. Since the StdPep approach is independent from these effects, we decided to use StdPep in our peptidomics analysis. We provided some more detail for the decision on StdPep in the Results section.
- b. Thank you very much for pointing the attention to the number of tryptic peptides. We were pretty impressed as even the % of #tryptic peptides, which represent a kind of intrinsic normalization of screening data, increased in ALS. The % of #tryptic peptides might help to monitor small changes of overall enzymatic activity during disease progression but also between different neurodegenerative

diseases thus contributing to a better covering of pathophysiological processes. However, this would expand the scope of this manuscript and therefore, remains to be investigated in the future studies, but hopefully we could show with that the hidden potential behind peptidomics global approach.

3. *I found it challenging to assess the biomarker authenticity at "one single time" point. While it might be challenging, assessing these markers from ASO-treated SOD1-ALS patients from different time points and comparing the biomarker changes and their correlations to the motor enhancement scores (ALSFRS) might provide the best evidence for verifying the biomarker panels in this study. Similar studies were performed before such as PMID: 37064776.*

Response

Thank you very much for this excellent suggestion. We agree with the reviewer that the investigation of our peptide panel in ASO-treated SOD1-patients would be a good example to evaluate their performance to monitor treatment effects. As also mentioned by the reviewer, it must be considered that these are very rare samples given the low prevalence of SOD1-ALS and the short time since the approval of tofersen which are not easily available. We think it would also be beyond the scope of this manuscript to also extensively evaluate the value of the peptide panel for the monitoring of treatment effects in ASO-treated SOD1-ALS patients. This must be investigated in future studies. To better emphasize the relevance of our peptides as ALS biomarker candidates, we included results from a neurodegeneration cohort consisting of patients with Alzheimer's disease (AD), behavioral variant of frontotemporal dementia (bvFTD) and Parkinson's disease (PD) which enables the evaluation of the specificity of the peptide changes for ALS. Nevertheless, we are well aware that further investigation is needed to explore the potential of the presented candidates and we stated this in the conclusion section.

Minor critics:

1. *Would it be possible for the authors to perform GO enrichment analysis to determine whether the identified dysregulated peptides in the ALS group are associated with specific biological processes?*

Response

The clustering analysis performed in STRING database identified two big clusters. Based on information for proteins, i.e., biological processes, molecular function, cellular component, disease-gene associations, and tissue expression obtained from the STRING database, clusters were named as metabolic and neurodegeneration cluster, thus covering the main pathophysiological processes characterizing ALS. The validation cohort reduced the most promising biomarker candidates to eight peptides derived from seven proteins, in which both clusters are represented by NFL, MAP1B and APOC1. The functions of seven parent proteins are discussed separately. Thus, we think our analyses provide information about the association of proteins with specific biological processes.

However, if the reviewer still sees a need for an additional analysis, we can include it.

2. In Figure 2G, the authors perform peptidomic screening analysis on the discovery cohort and observe a separation of ALS groups in the hierarchical clustering analysis. What is the primary factor driving this separation (e.g., disease subtype, severity, gender, age, etc.)? Additionally, it would be valuable to discuss potential differences in peptidomics across various ALS subtypes.

Response

The clustering analysis was performed only with the abundance of peptides found significantly regulated and patient specific data were not included. In our discovery cohort, only sporadic ALS patients were included and consequently a comparison between various ALS subtypes using screening data remains to be done in up-coming studies. However, we performed a comparison of ALS subgroups in the validation cohort with our selected peptide panel and could indeed identify subgroup-specific changes.

3. In Table 1, there are several suggestions as below:
 - a) I assume that genetic ALS (gALS) refers to familial ALS (fALS). In most literature, individuals with familial ALS who carry mutations in known ALS-associated genes are typically referred to as having fALS. Is there a specific reason for using "gALS" in this manuscript?
 - b) Although the authors separated the groups by gender, I was wondering about the female-to-male ratio within each genetic type under "gALS."
 - c) Providing abbreviations in the table helps readers understand the content. However, the order appears to follow an alphabetical arrangement rather than a logical flow. It may be more intuitive to present the abbreviations in the order they appear from row to column.
 - d) There should be a superscript "c" instead of "d."

Response

- a. ALS mutations might also appear sporadically without a family history. By using the term "genetic" ALS, we wanted to be more general covering this point since we don't know the family history of all of our genetic ALS cases.
 - b. 6F/6M in C9orf72 and 2F/9M in SOD1 mutation carriers. We added this information to the table.
 - c. Rearranged in the order as they appear from row to column.
 - d. The superscription of the third P value changed from d to c.
4. In Table 2, what is the condition of evaluating the data using a shorter LC gradient? How does it differ from the regular conditions, and how should the results be interpreted?

Response

We developed two different LC methods for the measurement of the eight peptide candidates, 30 minutes and 60 minutes total run time. We registered a good

signal of PENK peptide in a CSF pool sample for the 1to2 dilution stability test with 60min but not 30min method. In contrary, MAP1B peptides showed good signals for dilution stability with the shorter 30min method. From screening and label-free PRM data we expected in ALS PENK to be decreased and MAP1B increased. Therefore, we decided to go with the longer gradient which is in favor of the PENK peptide. For the MAP1B peptides, however, we included the 1to2 and 1to4 dilution stability test results obtained with the shorter method.

5. *There are some typos or inconsistent characters in the text and figures. Please review and correct them.*

Response

Thank you for this advice. We carefully reviewed the manuscript and corrected the typos and inconsistent characters.

Referee #3

Major critics:

1. *In this analysis, age- and sex-matched individuals without neurodegenerative diseases were used as controls. However, in clinical practice, ALS is often misdiagnosed or confused with other motor neuron diseases. Including patients with such conditions in the validation cohort could enhance the clinical translational potential of the study.*

Response

We here included non-neurodegenerative controls for comparison with ALS patients. This were patients that presented in the neurology clinic in Ulm for diagnostic work-up and not healthy individuals and thus a relevant patient population for differential diagnosis. We agree with the reviewer that in a clinical setting ALS mimics might be more relevant for differential diagnosis. However, this is an exploratory study to identify novel biomarker candidates and the use of non-neurodegenerative control samples is a common procedure here. Further studies must further clarify the exact differential diagnostic value. We adapted our final statement in the conclusion section with this point.

2. *The study identified eight peptides as potential biomarkers for ALS, derived from seven different proteins. I am particularly interested in whether these source proteins are present in the CSF. If they are not found in the CSF, it would further underscore the significance of this peptidomic analysis. Conversely, if these proteins are present in ALS CSF, do their levels follow a similar trend as the identified peptides? Additionally, the study mentions that the peptide derived from MYL1 exists in a trimethylated form. I am curious whether this modification occurs on the intact MYL1 protein or if it arises post-cleavage at the peptide level.*

Response

We would like to make clear that all peptide sequences are reported here for the first time as deregulated in ALS and biomarker candidates, including NFL peptide. The immunoassays commonly used to measure NFL in clinical samples targets other regions of NFL.

Yes, all proteins are present in CSF samples as reported in previous publications by us (Oeckl et al, Acta Neuropathologica 2020) and others (see references in the manuscript, not published yet for MYL1). To the best of our knowledge, we detected for the first time MYL1 in CSF sample, Protein and peptide changes were not necessarily similar and strongly depended on the peptide e.g. for MAP1B only two peptides were significantly changed in ALS whereas all others are not. This highlights the importance and added value of peptidomic analysis in CSF for biomarker research (page 12, line 270).

We identified two MYL1 peptides and the one which did not fulfill the quantification criteria has only one K residue more in its amino acid sequence. The detection of only peptides located in the MYL1 N-terminal extension is indicative for its selective detachment from the parent protein. The unmodified form of MYL1 peptides were not detected in peptidomics and we still do not know whether the MYL1 protein is methylated or not. We are working on this topic.

- 3. The manuscript states that "PENK was especially down-regulated in C9orf72 mutation carriers even compared to sALS", suggesting potential differences in the CSF peptidome between sALS and qALS patients. Conducting a comparative analysis of data from sALS and qALS patients at the early experimental stage, particularly during the screening peptidomic analysis, could help identify additional differentially expressed peptides between the two groups. Such findings may offer valuable insights into the etiological differences between sALS and qALS.*

Response

We agree with the reviewer that this is a very interesting question and our data indicate that peptidomics is a promising technique to uncover differences between ALS subtypes. However, in the discovery cohort, we included only sporadic ALS patients to reduce the variability during screening. Therefore, the comparison of ALS subgroups in the screening peptidomics experiment is not possible but must be performed in future studies.

- 4. The manuscript reports significant correlations among the levels of up-regulated peptides. I am curious whether this implies a functional or regulatory relationship among these peptides. Although the analysis indicates no significant correlation between individual peptide levels and FRS r, could an integrated analysis of multiple peptides reveal a potential association with FRS r?*

Response

One of the main disadvantages of global peptidomics is that data are discussed primarily from the perspective of parent proteins (Muqaku et al IJMS 2022). Therefore, it is difficult to make any conclusion about the function of these peptides without studies with cellular or animal models, although, for MYL1 N-terminal extension, where the MYL1 peptide is located, studies with animal model

indicated its relevant role during the interaction of Myosin II complex with actin as already mentioned in the manuscript. From the view of parent protein, NFL, MAP1B and MYL1 are either part or interact with cell cytoskeleton. NFL builds intermediate filaments, MAP1B stabilizes microtubules and MYL1 as part of Myosin II complex undergo interaction with actin filaments (microfilaments) during muscle contraction. All three proteins describe neurodegeneration and impaired muscle function as two key processes characterizing ALS and hence their correlation. Another well recognized process in ALS is the increased energy demand forcing metabolic alteration and we believe that this impacted the elevated APOC1 levels.

A better correlation with FRS was not achieved by combination of NFL and MAP1B peptides, those peptides correlating with FRS.

5. I also have a few suggestions regarding the presentation of data and figures:

(1) For correlation plots, such as Figures 1H and 4F, I recommend displaying the correlation coefficients directly on the scatter plots to enhance clarity. Additionally, in Figure 4F, I suggest grouping up-regulated and down-regulated peptides, as this would enhance readability and facilitate interpretation.

Response

The peptides are grouped into up-regulated and down-regulated as in boxplots. The size of the number will be too small and hardly readable if the correlation coefficients would be placed into the scatter plots. Hence, the initial presentation of correlation coefficients is kept.

(2) In the figure legends, the placement of panel labels is inconsistent, appearing before the legend in some cases and after it in others. I recommend standardizing the format for clarity and consistency.

Response

Corrected.

(3) Some abbreviations, such as FRS r , are used in the manuscript without providing their full definitions. I recommend defining all abbreviations upon their first occurrence for clarity.

Response

Corrected.

(4) Certain analyses are discussed in the manuscript without the corresponding data being presented. I suggest either including the relevant data or revising the text to remove these references (e.g., Result Section 4, Paragraph 3).

Response

Data and figures were included as supplementary material.

(5) I recommend reorganizing the Methods section to align with the order of the main text. This would improve readability and help readers better understand the role of each technique in the study.

Response

We thank the reviewer for this important remark. However, we are not sure if we fully understand the suggestion. The Methods section actually follows the order of the Results section in the main text, except for the patient cohorts and the statistical analysis, which are typically placed at the beginning and end of the Methods section, respectively.

11th Jun 2025

Dear Dr. Oeckl,

Thank you for the submission of your revised manuscript to EMBO Molecular Medicine. I am pleased to inform you that we will be able to accept your manuscript pending the following final amendments:

- 1) Please address the referee #2 point 1 by acknowledging the limitation of the study and point 2 by implementing the suggestion to expand discussion on the methodological optimizations and include comparisons with existing databases.
- 2) Please address all comments suggested by our data editors listed below:
 - o Data availability statement:
 1. Please note that the specific URL for PXD062419 dataset is not provided in the data availability statement.
 - o Figure legends:
 1. Please note that the exact p values are not provided in the legends of figures 2I, J; 3B, 4A-C; 6, EV2.
 2. Please note that the box plots need to be defined in terms of minima, maxima, centre, bounds of box and whiskers, and percentile in the legends of figures 2I, J; 3B.
 3. Please note that the box plots need to be defined in terms of minima, maxima and percentile in the legends of figures 4A-C; 6, EV2.
 4. Please note that information related to n is missing in the legends of figures 2I, J
- Correct order of manuscript sections: Abstract / Keywords / The Paper Explained / Introduction / Results / Discussion / Methods / Data Availability / Acknowledgements / Disclosure and Competing Interests Statement / References / Main Figure Legends / Tables / Expanded View Figure Legends.
- Limit keywords to max. 5.
- Add callouts for Figure 1A. Also, there is a callout for a Suppl. Table 1, please correct.
- In Methods, provide the statement that informed consent was obtained from all human subjects and confirm that the experiments conformed to the principles set out in the WMA Declaration of Helsinki and the Department of Health and Human Services Belmont Report.
- In Methods, add the following paragraph:

Graphics:

(some of the... OR Figure #... OR synopsis) Graphics were created with BioRender.com.

- Indicate in legends exact n and exact p values, not a range, along with the statistical test used. To keep the figures "clear" some authors found providing an Appendix table Sx with all exact p-values preferable. You are welcome to do this if you want to.
 - Please remove Reagents and Tools Table and uploaded it as a separate file. Structured Methods section includes Reagents and Tools Table followed by a Methods and Protocols section. More information on how to adhere to this format as well as downloadable templates (.docx) for the Reagents and Tools Table can be found in our author guidelines: <https://www.embopress.org/page/journal/17574684/authorguide#structuredmethods>
- An example of a paper with Structured Methods can be found here:
<https://www.embopress.org/doi/full/10.1038/s44320-024-00037-6#sec-4>
- Please use the following format to report the accession number of your data:

[data type]: [full name of the resource] [accession number/identifier] [(doi or URL or identifiers.org/DATABASE:ACCESSION)]

Please check "Author Guidelines" for more information.

<https://www.embopress.org/page/journal/17574684/authorguide#availabilityofpublishedmaterial>

- 3) Tables: Rename Table EV1 and EV2 to Dataset EV1 and EV2 and renumber Tables EV3-EV8 accordingly. Please update their callouts in the main manuscript text and upload tables as excel files directly, there is no need to ZIP them first.
- 4) Funding: Please merge it with "Acknowledgements". Also, please remove the reference to BioRender from the acknowledgements.
- 5) The Paper Explained: Please add it to the main manuscript text.
- 6) Synopsis:
 - Synopsis image: Please crop and resize the image to 550 px-wide x 300-600 pixels high and upload it as a high-resolution jpeg file. Make sure that text is readable and all the items in the image clearly visible.
 - Please check your synopsis text and image before submission with your revised manuscript. Please be aware that in the proof stage minor corrections only are allowed (e.g., typos).
- 7) As part of the EMBO Publications transparent editorial process initiative (see our Editorial at <http://embomolmed.embopress.org/content/2/9/329>), EMBO Molecular Medicine will publish online a Review Process File (RPF) to accompany accepted manuscripts. This file will be published in conjunction with your paper and will include the anonymous referee reports, your point-by-point response and all pertinent correspondence relating to the manuscript. Let us know whether you agree with the publication of the RPF and as here, if you want to remove or not any figures from it prior to publication. Please note that the Authors checklist will be published at the end of the RPF.

8) Please provide a point-by-point letter INCLUDING my comments as well as the reviewer's reports and your detailed responses (as Word file).

I look forward to reading a new revised version of your manuscript as soon as possible.

Yours sincerely,

Zeljko Durdevic

Zeljko Durdevic
Senior Editor
EMBO Molecular Medicine

*** Instructions to submit your revised manuscript ***

- 1) a .docx formatted version of the manuscript text (including Figure legends and tables)
- 2) Separate figure files*
- 3) supplemental information as Expanded View and/or Appendix. Please carefully check the authors guidelines for formatting Expanded view and Appendix figures and tables at <https://www.embopress.org/page/journal/17574684/authorguide#expandedview>
- 4) a letter INCLUDING the reviewer's reports and your detailed responses to their comments (as Word file).
- 5) The paper explained: EMBO Molecular Medicine articles are accompanied by a summary of the articles to emphasize the major findings in the paper and their medical implications for the non-specialist reader. Please provide a draft summary of your article highlighting
 - the medical issue you are addressing,
 - the results obtained and
 - their clinical impact.This may be edited to ensure that readers understand the significance and context of the research. Please refer to any of our published articles for an example.
- 6) Author contributions: the contribution of every author must be detailed in a separate section.
- 7) EMBO Molecular Medicine now requires a complete author checklist (<https://www.embopress.org/page/journal/17574684/authorguide>) to be submitted with all revised manuscripts. Please use the checklist as guideline for the sort of information we need WITHIN the manuscript. The checklist should only be filled with page numbers where the information can be found. This is particularly important for animal reporting, antibody dilutions (missing) and exact values and n that should be indicated instead of a range.
- 8) Every published paper now includes a 'Synopsis' to further enhance discoverability. Synopses are displayed on the journal

webpage and are freely accessible to all readers. They include a short stand first (maximum of 300 characters, including space) as well as 2-5 one sentence bullet points that summarise the paper. Please write the bullet points to summarise the key NEW findings. They should be designed to be complementary to the abstract - i.e. not repeat the same text. We encourage inclusion of key acronyms and quantitative information (maximum of 30 words / bullet point). Please use the passive voice. Please attach these in a separate file or send them by email, we will incorporate them accordingly.

You are also welcome to suggest a striking image or visual abstract to illustrate your article. If you do please provide a jpeg file 550 px-wide x 300-600px high.

9) A Conflict of Interest statement should be provided in the main text

10) Please note that we now mandate that all corresponding authors list an ORCID digital identifier. This takes <90 seconds to complete. We encourage all authors to supply an ORCID identifier, which will be linked to their name for unambiguous name identification.

Currently, our records indicate that the ORCID for your account is 0000-0002-7652-7023.

Link Not Available

11) Include a Reagents and Tools Table as part of the Methods section, which can be downloaded from our author guidelines (<https://www.embopress.org/page/journal/17574684/authorguide#structuredmethods>)

Photos 400-800 DPI

*Additional important information regarding figures and illustrations can be found at

<https://bit.ly/EMBOPressFigurePreparationGuideline>. See also figure legend preparation guidelines:

<https://www.embopress.org/page/journal/17574684/authorguide#figureformat>

***** Reviewer's comments *****

Referee #2 (Remarks for Author):

The manuscript presents a valuable investigation into peptide biomarkers for ALS using peptidomics analysis of CSF. The authors have made several improvements in response to the reviewers' earlier concerns, including the addition of a neurodegenerative disease cohort and clarification of their methodological framework. However, a few critical points remain inadequately addressed and should be considered before publication:

1. Although the authors state that the peptides identified in this study behave independently from their parent proteins and are reported for the first time, the manuscript lacks direct experimental validation to substantiate this claim, particularly regarding the functional relevance of these biomarkers. It remains unclear whether these candidate biomarkers can reliably predict disease progression in sporadic ALS patients or be effectively used for staging and longitudinal monitoring.

2. It would be beneficial to expand on the methodological optimizations in the Discussion section and include comparisons with existing databases to strengthen the claim that these modifications improve the clarity and reliability of novel peptide discovery in CSF samples from patients.

Referee #3 (Remarks for Author):

I don't have further question.

Referee #2

The manuscript presents a valuable investigation into peptide biomarkers for ALS using peptidomics analysis of CSF. The authors have made several improvements in response to the reviewers' earlier concerns, including the addition of a neurodegenerative disease cohort and clarification of their methodological framework. However, a few critical points remain inadequately addressed and should be considered before publication:

1. Although the authors state that the peptides identified in this study behave independently from their parent proteins and are reported for the first time, the manuscript lacks direct experimental validation to substantiate this claim, particularly regarding the functional relevance of these biomarkers. It remains unclear whether these candidate biomarkers can reliably predict disease progression in sporadic ALS patients or be effectively used for staging and longitudinal monitoring.

Response

We thank the reviewer for the relevant remarks. Following the editor suggestion, a paragraph has been added in the end of the Discussion: The main limitations of this study are the lack of direct experimental validation of independent behavior of peptides from their parent proteins and unclarity whether these candidate biomarkers can reliably predict disease progression or be effectively used for staging and longitudinal monitoring (page 19, line 480).

2. It would be beneficial to expand on the methodological optimizations in the Discussion section and include comparisons with existing databases to strengthen the claim that these modifications improve the clarity and reliability of novel peptide discovery in CSF samples from patients.

Response

In the Results and Discussion sections the following statements have been added: A search of the peptidomics data against a peptide library created from the peptidatlas FASTA file from peptipedia 2.0 identified only 217 additional peptides with unaltered abundance in ALS compared to Con (page 7, line 168). To date, the highest number of peptides identified in CSF of neurodegenerative disease patients is 18 031, obtained through extensive sample fractionation. We identified 33 605 peptides in CSF samples (page 18, line 442). The high number of peptide IDs was achieved through the use of state-of-the-art MS instrumentation, improvements in software packages, and systematic optimization of the sample preparation protocol and LC-MS parameters specifically for peptidomic analysis of CSF samples. Searching the peptidomics data against peptide database did not significantly improved the number of identified peptides, demonstrating the comprehensiveness of our initial search against protein database (pages 19, line 449).

Referee #3

I don't have further question.

Response

We thank the reviewer for the valuable contribution to our manuscript.

3rd Jul 2025

Dear Dr. Oeckl,

We are pleased to inform you that your manuscript is accepted for publication and is now being sent to our publisher to be included in the next available issue of EMBO Molecular Medicine.

Zeljko Durdevic
Senior Editor
EMBO Molecular Medicine
